# Identification of atrial fibrillation associated genes and functional non-coding variants

Antoinette F. van Ouwerkerk [1], Fernanda M. Bosada[1,15], Karel van Duijvenboden[1,15], Matthew C. Hill [2], Lindsey E. Montefiori[3], Koen T. Scholman[1], Jia Liu[4,5,6], Antoine A.F. de Vries[4,6], Bastiaan J. Boukens[1], Patrick T. Ellinor [7,8,9], Marie José T.H. Goumans [10], Igor R. Efimov [11], Marcelo A. Nobrega[3], Phil Barnett[1], James F. Martin[2,12,13,14] & Vincent M. Christoffels[1]*

Disease-associated genetic variants that lie in non-coding regions found by genome-wide association studies are thought to alter the functionality of transcription regulatory elements and target gene expression. To uncover causal genetic variants, variant regulatory elements and their target genes, here we cross-reference human transcriptomic, epigenomic and chromatin conformation datasets. Of 104 genetic variant regions associated with atrial fibrillation candidate target genes are prioritized. We optimize EMERGE enhancer prediction and use accessible chromatin profiles of human atrial cardiomyocytes to more accurately predict cardiac regulatory elements and identify hundreds of sub-threshold variants that co-localize with regulatory elements. Removal of mouse homologues of atrial fibrillation-associated regions in vivo uncovers a distal regulatory region involved in *Gja1* (Cx43) expression. Our analyses provide a shortlist of genes likely affected by atrial fibrillation-associated variants and provide variant regulatory elements in each region that link genetic variation and target gene regulation, helping to focus future investigations.

[1] Department of Medical Biology, Amsterdam University Medical Centers, Academic Medical Center, 1105 AZ Amsterdam, The Netherlands. [2] Program in Developmental Biology, Baylor College of Medicine, Houston, TX 77030, USA. [3] Department of Human Genetics, The University of Chicago, Chicago, USA. [4] Department of Cardiology, Leiden University Medical Center, Albinusdreef 2, 2333 ZA Leiden, The Netherlands. [5] Department of Cell Biology and Genetics, Center for Anti-ageing and Regenerative Medicine, Shenzhen Key Laboratory for Anti-ageing and Regenerative Medicine, Shenzhen University Medical School, Shenzhen University, Nanhai Ave, 3688 Shenzhen, China. [6] Netherlands Heart Institute, Holland Heart House, Moreelsepark 1, 3511 EP Utrecht, The Netherlands. [7] Cardiovascular Disease Initiative, Broad Institute of MIT and Harvard, Cambridge, MA, USA. [8] Cardiovasular Research Center, Massachusetts General Hospital, Charlestown, MA, USA. [9] Cardiac Arrhythmia Service, Massachusetts General Hospital, Boston, MA, USA. [10] Department of Cell and Chemical Biology, Leiden University Medical Center, Einthovenweg 20, 2333 ZC Leiden, The Netherlands. [11] Department of Biomedical Engineering, George Washington University, Washington, DC, USA. [12] Department of Molecular Physiology and Biophysics, Baylor College of Medicine, Houston, TX 77030, USA. [13] Texas Heart Institute, Houston, TX 77030, USA. [14] Cardiovascular Research Institute, Baylor College of Medicine, Houston, TX 77030, USA. [15] These authors contributed equally: Fernanda M. Bosada, Karel van Duijvenboden. *email: v.m.christoffels@amsterdamumc.nl

Atrial fibrillation (AF) is the most common arrhythmia, affecting over 34 million people worldwide[1]. AF can lead to life-threatening complications such as stroke, heart failure, dementia or death, and has a clear public health and economic relevance[2]. In recent years, genome-wide association studies (GWAS) have identified over 100 genetic loci associated with AF[3–12]. Although for some loci the potential mechanisms through which the AF GWAS loci add to the risk of AF have been identified, the majority have remained largely elusive[13].

A significant association between AF variants and differential expression of several target genes has been shown in several expression quantitative trait loci (eQTL) studies[10,12,14–17]. However, most AF risk loci do not have known eQTLs, and the limited availability and quantity of informative human tissues also limits the power of eQTL studies. To gain insight into the causal relation between AF variants, genes and AF risk, it is crucial to determine the potential target genes of each AF-associated locus and to identify the effect-causing SNP among the many variants associated with AF[13].

Interestingly, each AF-associated variant region is comprised of many associated SNPs and the vast majority of these variants are located in non-coding regions. For several common diseases, non-coding GWAS risk variants were found to lie within tissue-specific regulatory elements (REs), likely influencing their function[18–21]. Risk-associated variants in REs have been associated with altered transcription of the REs' target genes, suggesting a mechanistic link between variant and altered gene expression underlying disease[22–26]. REs regulate target genes through transcription factor (TF) binding and DNA looping, bringing the REs and target gene promoter(s) in close physical proximity in three dimensional space[27,28]. The majority of functional RE-target promoter interactions occur within the same topologically associating domain (TADs);[29–31] disruption of TAD boundaries can cause REs to activate genes in neighboring TADs (reviewed in refs. [32–34]). Nevertheless, TAD structure variability and inter-TAD interactions have been observed as well[35]. Moreover, recent studies showed that promoter interactions in differentiated cardiomyocytes (CMs) can identify non-coding DNA that is enriched for active REs that interact with cardiac disease relevant target genes[36,37]. However, physical proximity between REs and promoters—as assessed by high resolution chromosome conformation capture technologies[32]—is required, but not sufficient, and not all interactions are detected by the conformation capture assays[28,32,36]. Therefore, determining which specific gene(s) is/are regulated by a variant-affected RE remains a challenge.

To prioritize candidate target genes of non-coding AF-associated variant regions, here we compile and cross reference available and novel human and mouse transcriptomic, epigenomic, and chromatin conformation datasets (Supplementary Fig. 1). To examine the region's impact on putative target gene expression, we delete homologues of three of the AF-associated regions in the mouse using genome editing. We identify potential cardiac REs in each AF-associated region by ATAC-seq of CM nuclei from human atrial tissue and by deploying EMERGE enhancer prediction tool[38] that we retrain for more accurate prediction of human cardiac REs. Our analysis provides a rich resource of genes potentially associated with AF, variant REs and detailed epigenomic and transcriptomic signatures of atrial CMs that can serve to guide functional studies of the genetic predisposition for AF.

## Results

**Potential target genes of AF-associated variant regions.** We set out to prioritize the target genes that may be affected by the AF-associated genetic variants in the 104 associated loci[3–12] using the

following assumptions: (1) some non-coding AF-associated variants in a variant region are positioned in REs and affect the expression of their target gene(s), (2) such variant REs and the promoters of the genes they control (target genes) are in close physical proximity, and (3) target genes of variant REs are expressed in atrial tissue (analysis summarized in Supplementary Fig. 1a and Supplementary Fig. 2a). Because any AF-associated variant close to the lead SNP could be affecting RE function, we set out to define the genomic range of each AF-associated variant region. We tested various different criteria (SNPs in linkage disequilibrium, association p-value cut offs) to define the variant regions (Supplementary Fig. 3). Using the widely accepted genome-wide threshold of $p < 1 \times 10^{-8}$ for AF association of variants resulted in variant regions of high confidence and practical sizes[12,39]. These regions most likely contain the variant REs that govern candidate target gene expression. Sub-threshold SNPs (association $p < 1 \times 10^{-4}$) can affect RE activity and are highly likely to represent true disease risk loci[40], which may be excluded when they flank the variant region borders as defined by these stringent criteria. Therefore, we extended the variant regions in order not to lose potentially functional variants at the margins of each locus by marking the last variant with $p < 1 \times 10^{-6}$ at either side beyond the $p < 1 \times 10^{-8}$ boundaries, leading to a modest increase in variant regions of on average 69 kb (22%) at each boundary (Supplementary Table 1).

RE-target promoter interactions over distances of up to 1.9 Mb have been reported[41]. Therefore, to identify genes within reach of potential variant REs, we identified all genes 1.9 Mb upstream to 1.9 Mb downstream of the lead SNPs as the potential target genes per locus (Supplementary Data 1). However, RE activity is largely limited to genes that fall within the same TAD[32–34]. Since TAD structures are largely tissue independent, we determined the TAD of the variant region by interrogating available Hi-C data for each of the 104 AF-associated loci in our study[42]. We indicated for each gene 1.9 Mb up- or downstream of the lead variant whether or not it was within the same TAD as the variant region (Supplementary Data 1). On average, the TADs had a length of 1 Mb, with an average of 10 genes per TAD (Table 1). Next, we assessed which gene promoters are contacted by the variant regions by interrogating PCHi-C data derived from human induced pluripotent stem cell-derived (hiPSC)-CMs[36]. The PCHi-C dataset was generated using a 4 bp cutter, generating fragments of smaller size and increased resolution of underlying RE compared to data generated by a 6 bp cutter. Each gene that showed a promoter contact with a variant region was marked (Supplementary Data 1).

The most common sites of AF initiation are located in the left atrial posterior wall[43]. Thus, we performed RNA-seq on the free wall of three matching human whole tissue left atrial dorsal walls, and also compared the profiles to those obtained from right atrial free walls. Hierarchical clustering showed that left atrium (LA) and right atrium (RA) samples, respectively, clustered together (Fig. 1c). In total, 5510 genes showed expression above 10 reads per Kb (RPK) after normalization (Fig. 1a). We found that *BMP10* and *SMYD2* were highly enriched in the RA and *PITX2* in the LA, as expected (Fig. 1b). Our data is consistent with RNA-seq datasets derived from atrial appendages[44,45] and free wall[46]. Furthermore, we observed striking left-specific expression of *MYOT*, and right-specific expression of *AFF2*. Global differential expression analysis indicated that 107 genes were differentially expressed ($-1 > L2FC > 1$, $p$adj $< 0.05$), of which 67 were enriched in the RA and 40 in the LA (Supplementary Data 2).

Whole atrial tissue contains several different cell types, and the resulting transcriptomes reflect this heterogeneous tissue composition. Although there is a role for non-CMs in AF[47], it is well-established that CMs can initiate and propagate

**Table 1 Scoring calculation method AF target genes**

| Type | Dataset/technique | Signal | Weight in score | Max score |
|---|---|---|---|---|
| Conformation | TAD | 0–1 | 2× | 2 |
| Conformation | PCHi-C | 0–1 | 3× | 3 |
| Expression | RNAseq adult whole tissue left atria | 0–2 | 1× | 2 |
| Expression | RNAseq adult whole tissue right atria | 0–2 | 1× | 2 |
| Expression | RNAseq adult left atrial cardiomyocytes | 0–2 | 1× | 2 |
| Expression | RNAseq fetal left whole tissue atria | 0–2 | 1× | 2 |
| Expression | RNAseq fetal right whole tissue atria | 0–2 | 1× | 2 |
| eQTL | eQTL | 0–2 | 4× | 8 |
| | | | | 23 |

Range of signal of each dataset, as well as the weight given to each dataset used for the calculation of target gene analysis is given

arrhythmogenicity[48,49]. Therefore, we wanted to assess which genes are enriched in CMs. To this end, we isolated CM nuclei of two human LA dorsal walls using PCM1-specific antibodies that selectively bind to CM nuclear membranes[50], and performed RNA-seq on these two samples. Differential expression analysis of the datasets revealed that out of 5580 differentially expressed genes, 2488 were significantly enriched in whole tissue and, including established fibroblast and endothelial markers, 3092 were enriched in PCM1$^+$ CM nuclei, including established CM-specific markers (Fig. 1d, e). We inventoried the whole tissue and CM-specific atrial expression of all genes 1.9 Mb up- and downstream of each AF-associated locus (Supplementary Data1) to allow comparison of expression for the assessment of potential AF target genes.

Changes in expression leading to AF may start to develop before birth. To include developmental expression in our analysis, we assessed the transcriptomes of whole tissue human fetal LA ($n = 2$, 14 and 22 weeks) and RA ($n = 2$, 14 and 22 weeks). When comparing fetal and adult atrial expression, 3976 genes were significantly enriched in adult, and 6155 genes in fetal tissue (Supplementary Data 2). Known fetal genes such as *NKX2-5* and *HAND2*[51] were highly expressed in fetal atria, while *PLN* and *PRRX1* were enriched in adult tissue (Fig. 1f, Supplementary Data 2). Again, we added the interaction scores per gene for each AF locus (Supplementary Data 1). Furthermore, we analyzed the top 3,000 differentially expressed genes ($p$-value < 0.05) for both datasets. GO term analysis using DAVID showed that the fetal transcriptome was enriched for terms such as nucleosome assembly, cardiac muscle tissue morphogenesis and cardiac muscle development, reflecting the developmental state (Fig. 1g). The genes differentially expressed in adult atria were enriched for GO terms such as muscle contraction and regulation of heart rate (Fig. 1h).

Next, we inventoried known eQTL data reported in literature[10,12,14–17]. In 36 loci, a risk SNP or SNPs in high LD with the risk SNP were associated with variation in the gene expression of a gene in human RA appendage, left ventricle or skeletal muscle tissue. eQTL thus indicates an association between risk SNPs and changes in expression of a particular gene. Therefore, we included these data in the score calculation of the candidate target gene probability assessment by listing which genes 1.9 Mb up- and downstream of the AF lead SNPs have been linked to AF via eQTL (Supplementary Data 1).

A custom scoring system was used to determine the likelihood of each gene in the 3.8 Mb window centered on the lead SNP (1.9 Mb up and downstream) (Supplementary Data 1) being a functional target gene of the variant locus. The weight of each of these genes increased if the gene (1) and the variant region fall within the same TAD, (2) shows interaction with an AF-associated variant region (PCHi-C), (3) is expressed in adult human atria (RNA-seq of LA and RA), (4) is expressed in CMs of

adult human atria, (5) is expressed during development (fetal atria), and (6) has a known eQTL. After normalization using reference genes (*RPL32*, *RPL4* and *H2AF2*[52]) for the human RNA-seq datasets, the median expression was set as the boundary for what we consider expression (score 1), and the third quartile as boundary above which expression was considered high (score 2) (Supplementary Fig. 4).

A summary of all the genes within the 1.9 Mb range of the AF loci and their scores for PCHi-C and RNA-seq is provided in Supplementary Data 1, and a summary of the top scoring gene per locus in Supplementary Table 2. We set the threshold above which we consider a gene a potential target gene at 11 in order to remove genes that are expressed (maximal score of 10) but do not score on any of the other criteria. Furthermore, genes that are expressed and show close proximity to the variant region outside of the TAD are included using this score. Out of 3,304 genes in the 1.9 Mb up- and downstream of the lead SNPs of the 104 loci, 264 genes had a score above the threshold of 11, which is a reduction of 93% (Fig. 2a). Within the TADs there are 1040 genes, of which only 249 have a score of 11 or above (24%). For all analyses we removed duplicates of genes that are within the 1.9 Mb range of two different loci to give an accurate representation. GO term analysis on this selection of genes returned a significant enrichment of genes involved in cardiac conduction, force of contraction, and heart development (Fig. 2b, c). A summary of eight of the most highly associated loci is presented in Fig. 2d, showing that for some loci only a few genes reach the threshold.

In Fig. 3a an example locus is shown, with score calculation in Fig. 3e. Genes just outside the TAD are highly expressed in the heart (*CLK1*, *NDUFB3*, *CFLAR*), but do not show close proximity to the variant region and are therefore unlikely to be regulated by variant REs. *SGOL2* lies within the TAD and shows close proximity to the variant region, but is not expressed in the heart and is therefore unlikely to be involved in AF via a cardiac mechanism. No eQTLs are known for AF with any of the genes in this locus. The *SPATS2L* gene lies within the TAD of the variant region, its promoter shows close proximity to the variant region, the gene is highly expressed in adult atria, and expressed in LA CMs and in fetal atria. Concluding, this is the only gene with a score above 11 in this locus, and therefore the most likely variant region target gene involved in AF.

Having identified 264 potential AF target genes, we re-analyzed the RNA-seq data specifically for these genes to further detail their spatial, temporal, and cell type-specific cardiac expression patterns (Fig. 4). Comparing LA and RA, only 5 AF target genes were differentially expressed; *PITX2*, *MYOT* and *RPL3L* in the LA and *HCN4* and *FMOD* in the RA (Fig. 4a). When comparing LA whole tissue and CM, 111 genes were differentially expressed, of which 57 show higher expression in the CMs (Fig. 4b). In CMs, the highest degree of differential expression (L2FC 4.1) was seen

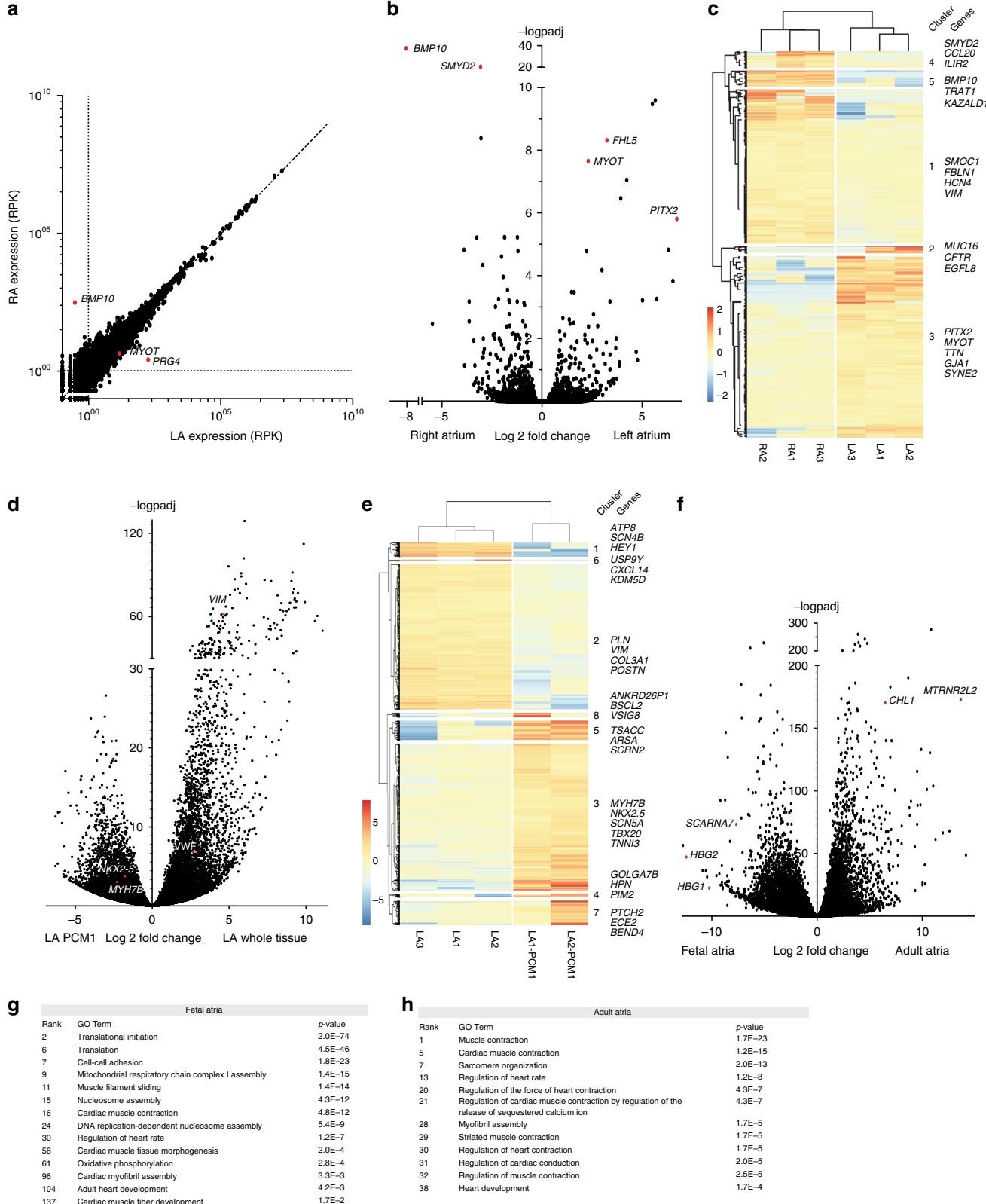

**Fig. 1** RNA-seq of human adult and fetal atria. **a** MA plot of left atrial and right atrial whole tissue expression (RPK). A total of 15,239 genes with RPKs >1 are detected in both samples. **b** Volcano plot analysis of left versus right adult atrial expression. **c** Heatmap of left and right atria ($n = 3$) shows that these tissues cluster together. **d** Volcano plot of adult left atrial tissue versus PCM1-sorted left atrial nuclei (i.e., CM nuclei). **e** Heatmap of whole tissue ($n = 3$) and PCM1-enriched left atria ($n = 2$). **f** Volcano plot analysis of adult ($n = 3$) versus fetal ($n = 2$) atrial expression. **g** GO terms of top 500 expressed genes in fetal atria with rank and *p*-value. **h** GO terms of top 500 expressed genes in adult atria with rank and *p*-value. *p*-values were calculated using DAVID (based on Fischer-exact test) and Benjamini-Hochberg corrected. RPK reads per kilo base, PCM1 pericentriolar material 1, GO Gene Ontology

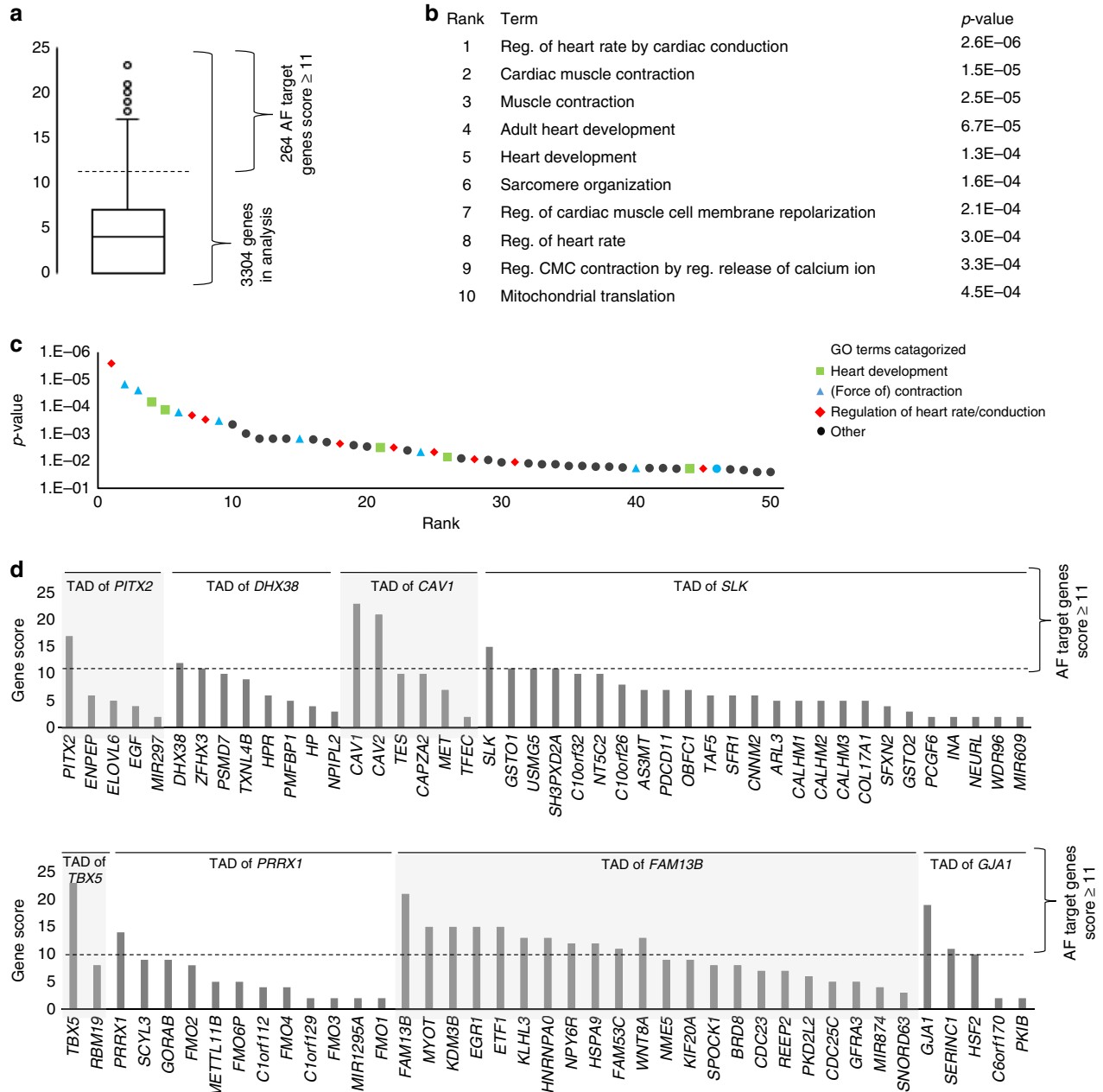

**Fig. 2** Scoring and target genes. **a** Boxplot showing distribution of all genes included in this study. **b** Top 10 functional GO terms for all genes with score > 11 and p-value of enrichment. **c** Top 50 functional GO terms categorized. **d** Summary of potential AF target genes with a score > 11. Locus number, nearest gene, and target genes per locus are shown, with their scores. GO Gene ontology, AF atrial fibrillation

for *PLN*. Between fetal and adult atria, differential expression was found for 175 genes, of which 97 are more abundant in the adult atria (Fig. 4c, d). For example, *ZFHX3*, *TTN,* and *MYOT* are more abundantly expressed in adult atria, while *RPS21*, *EFNA1*, and *HAND2* transcripts are more abundant in the fetal atria.

**Determining targets of AF-associated variant regions in vivo.** As a proof-of-concept, we used CRISPR/Cas9 genome editing to delete the mouse homologue of one of the AF risk loci on chromosome 6q22.31, around 680 Kbp downstream of *Gja1* in mice (Fig. 5a). A mouse line was generated carrying a germline deletion of 40 Kb corresponding to a 28-Kb region in the human genome (Fig. 5b, c). Homozygous mutant mice bred normally and did not show any visible developmental or anatomical

phenotype. Atria and brain cortexes were isolated from postnatal day 21 (P21) wildtype and homozygous deletion mice for analysis. The genes of interest were *Gja1* and *Serinc1* with score > 11 within the TAD (Fig. 5d). *GJA1* encodes connexin 43 (Cx43), the major gap junctional protein in the heart involved in cell-cell coupling[53] and impulse propagation[54]. Furthermore, because of its well-known function in the heart, we tested the effect of the deletion on the transcription of *Pln*[55], even though it lies in a different (neighboring) TAD. In the atria, *Gja1* was selectively reduced (p-value = 0.006) in expression as a result of the homozygous deletion (Fig. 5e, g). Expression levels of *Gja1* or the other genes selected was unaffected in brain tissue of homozygous mutant (Fig. 5f, h). We conclude that the 40-Kb region contains one or multiple REs that regulate expression of *Gja1* in a cardiac-specific manner.

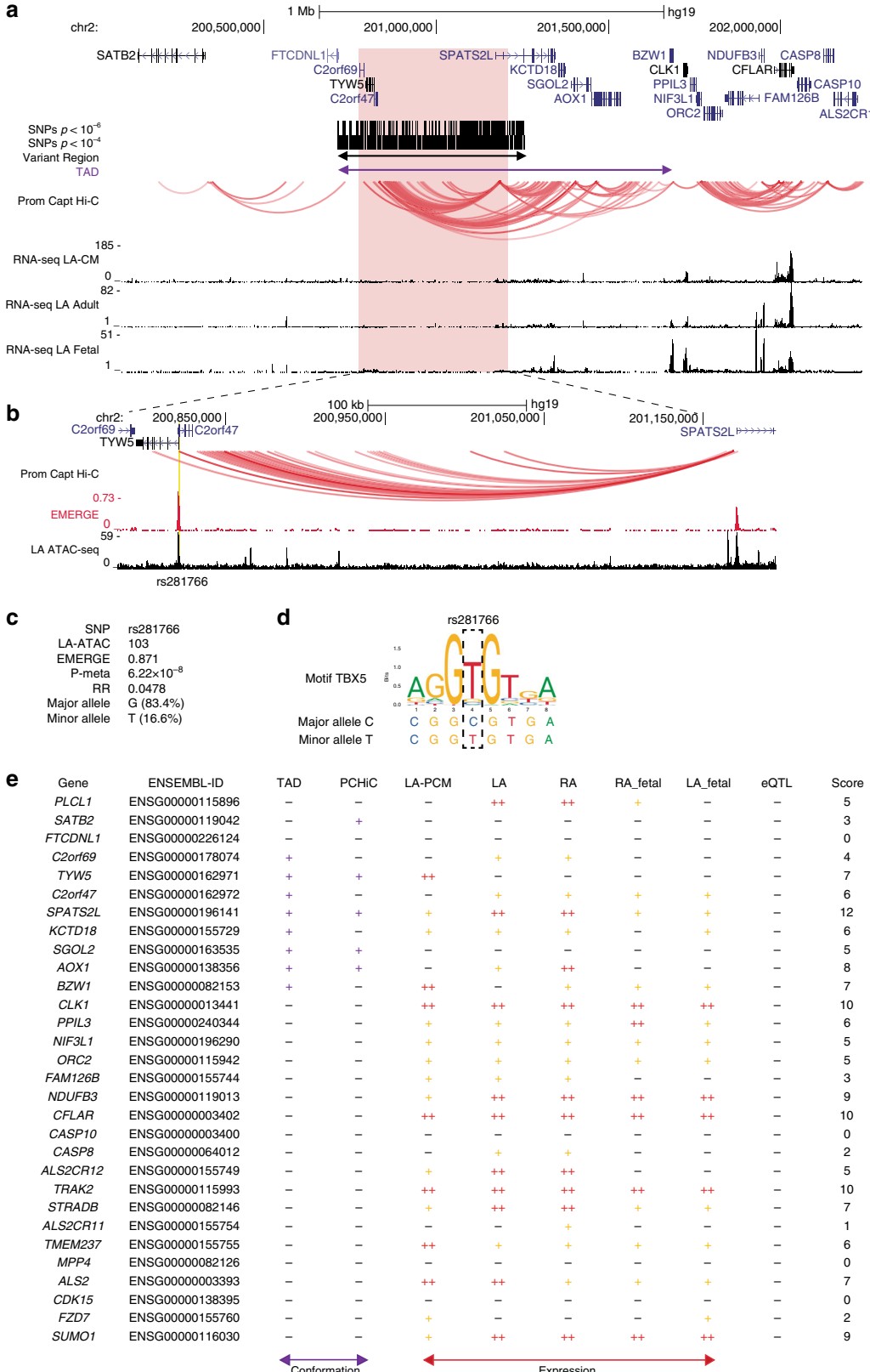

**Fig. 3** Target genes and variant enhancer of *SPATS2L* locus. **a** *SPATS2L* locus showing TAD, variant region, (sub)threshold variants, CM PCHi-C, and RNA-seq. **b** Zoom of part of the variant region, showing PCHi-C and epigenetic datasets EMERGE and LA ATAC-seq. rs281766 is highlighted in yellow. **c** Information of variant rs281766, **d** TBX5 recognition motif and minor and major alleles of variant rs281766. **e** Table showing PCHi-C interaction with variant regions, expression per gene and final scores for the analyzed genes. TAD Topologically Associated Domain, CM cardiomyocyte, PCHi-C promoter capture Hi-C

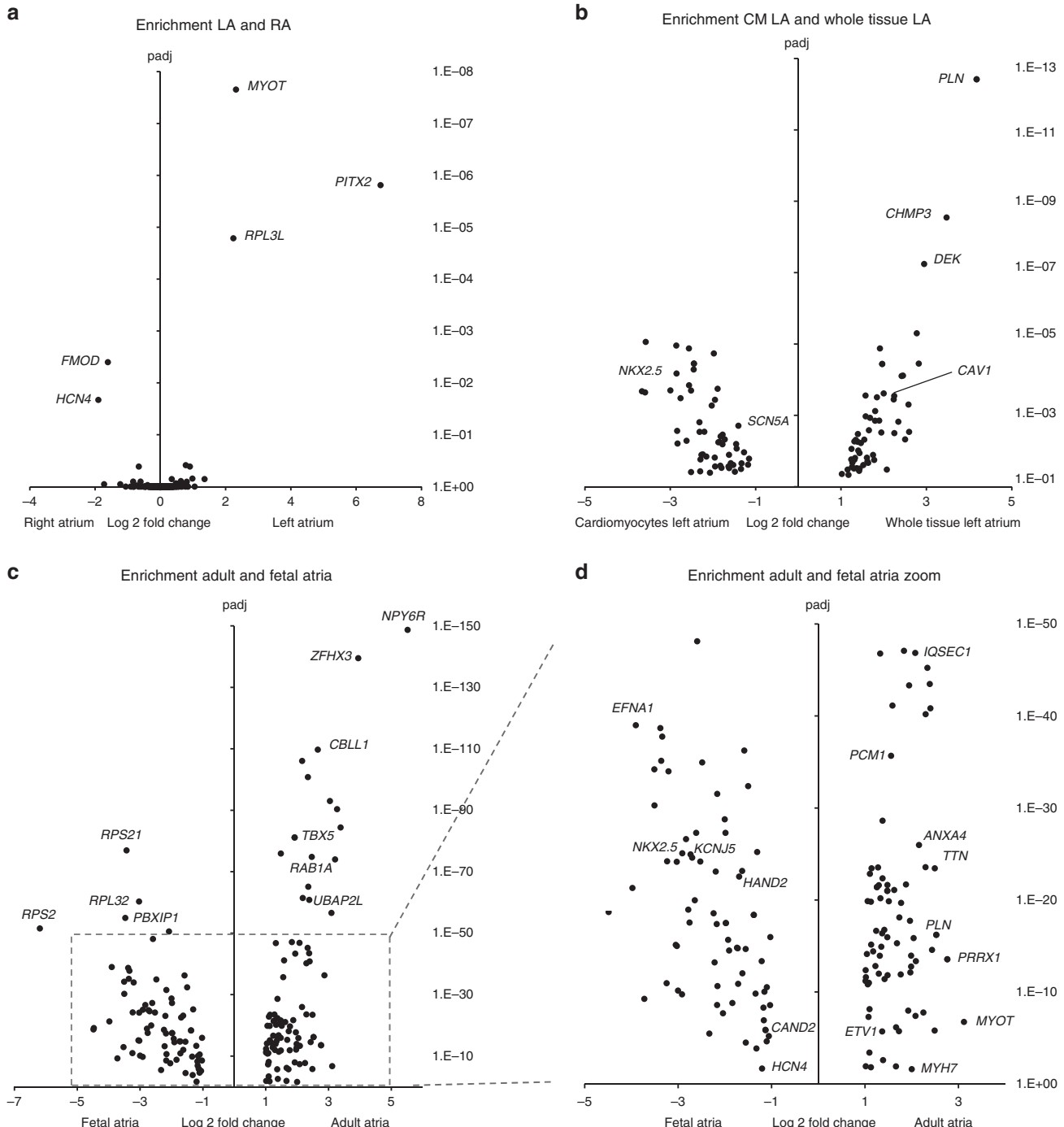

**Fig. 4** RNA-seq enrichment of AF target genes. Enrichment of expression of AF target genes (score > 11, −1 > L2FC > 1, padj < 0.05), comparing **a** enrichment of expression of LA and RA, **b** CM and whole tissue, **c** fetal and adult atria, and **d** zoom of fetal and adult atria. *p*-values are calculated with DESeq2 package (based on Wald-test) and corrected for multiple testing using false discovery rate method of Benjamini-Hochberg. Source data of all panels are provided as Source Data files. L2FC Log2-fold change, *p*adj adjusted *p*-value, CM cardiomyocyte, LA left atria, RA right atria

We utilized the same approach to test whether candidate gene expression is altered in the absence of an AF variant region in the 1q21 locus. We generated two independent mouse lines with an 85 Kb deletion including the promoter and first exon of *Kcnn3* (Supplementary Fig. 5a, b), thus providing a null allele and a potential enhancer deletion. The TAD containing the AF variants in this locus encompasses a large number of candidate target genes (Supplementary Data 1). Homozygous mutant animals exhibit loss of *Kcnn3* expression in atria and ventricles and a modest but significant decrease in expression of candidate genes

(score > 11) *Adar, and Shc1* in atria, of *Pmvk* and *Flad1* (score < 11) and of *Adar* and *Pmvk* in the ventricles and brain (Supplementary Fig. 5c, d), without a visible developmental or anatomical phenotype.

A third locus we examined is 16q22. The most highly AF-associated risk SNPs in this locus are found in the first intron of *ZFHX3*, which codes for a TF that plays key roles in development and in adult tissues[56,57]. To determine whether the AF-associated region has regulatory potential, we generated two independent mouse lines with a 33 Kb homozygous deletion in the first intron

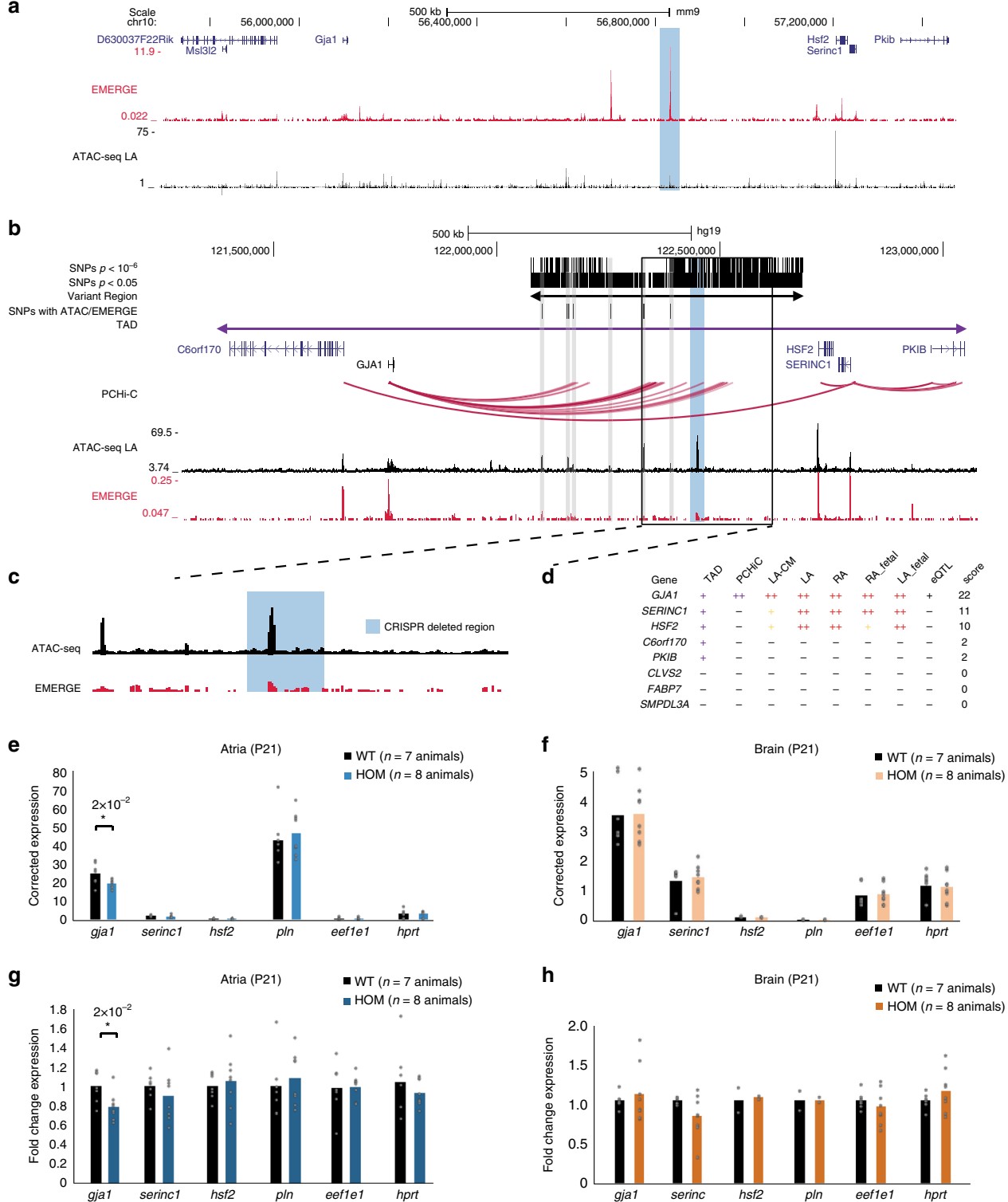

**Fig. 5** Variant region deletion shows distal regulatory element downstream of *Gja1* in mouse. **a** Mouse region homologous to human AF associated *GJA1* locus, showing gene, EMERGE and ATAC-seq tracks. Highlighted region is the 40 Kb deletion. **b** Browser view of human GJA1 locus, showing AF associated (subthreshold) SNPs in variant region (VR), TAD, genes, cardiac PCHi-C, ATAC-seq and EMERGE. Human homologous deleted region (highlighted). **c** Zoom of the human homologous deleted region (highlighted.) **d** Expression and PCHi-C values per gene in the TAD. **e** Corrected expression of genes as assessed by qPCR with *Hprt* and *Eef1e1* as reference genes in atria and **f** brain. **g** Expression fold change of *Gja1*$^{RE1/RE1}$ (HOM) vs wild-type (WT) littermates as a negative control of P21 atria and **h** brain cortex. Source data of 5**e**–**h** are provided as Source Data files. Error bars represent SD, *$p < 0.05$, **$p < 0.01$, and ***$p < 0.001$ (two-tailed Student's *t*-test). kb kilobase, SNP single nucleotide polymorphism, TAD topologically associated domain, PCHi-C promoter capture Hi-C, ATAC-seq assay for transposase-accessible chromatin sequencing, qPCR quantitative polymerase chain reaction, P21 postnatal day 21, SD standard deviation

of *Zfhx3* (Supplementary Fig. 6a, b). We observed no significant changes in the transcript levels of any of the selected genes, including *Zfhx3*, in P21 atria, ventricles, and brains or E15.5 hearts and hind limbs of homozygous mutant mice when compared to wild-type littermates (Supplementary Fig. 6c–f, g).

**Identification of cardiac REs and variant REs**. To identify REs within variant regions, we analyzed a chromatin accessibility dataset as well as an RE prediction tool that integrates different epigenetic datasets (EMERGE). We assessed the chromatin accessibility by interrogating an ATAC-seq dataset of human LA free wall CMs ($n = 11$) (MCH, JFM, unpublished). Within all variant regions, a total of 1206 ATAC-seq peaks were called, representing potential REs (Supplementary Data 3). Furthermore, we performed motif enrichment analysis on the ATAC-seq signals and found enrichment of binding motifs of established cardiac TFs and CTCF (Fig. 6a, Supplementary Table 3). To further validate the data quality and motif analysis we implemented the esATAC pipeline[58] and identified well-defined footprints of CTCF and MEF2A binding motifs in the ATAC-seq data set (Supplementary Fig. 7). As a second approach to map REs we used the EMERGE tool[38]. We first set out to improve the previously predicted heart enhancer catalogue. Seventy epigenetic data sets (ChIP-seq, ATAC-seq) relevant for human heart tissue were integrated and merged with the EMERGE algorithm, of which 53 were recent and therefore not present in the original EMERGE heart prediction (Supplementary Table 4)[38]. Furthermore, the functionally validated human heart enhancers[59] used to train EMERGE were curated because we observed that 17 of 126 reported cardiac enhancers were incorrectly annotated or weakly active in the heart (Supplementary Fig. 8a). Furthermore, we removed 25 enhancers active in the embryonic outflow tract (OFT), as they are likely active in neural crest rather than in myocardium. In total, we selected 84 validated, and correctly annotated enhancers to train EMERGE (Supplementary Fig. 8b). This resulted in an overall predicted heart enhancer catalogue with a prediction superior to the version published previously (Fig. 6b, c)[38]. Here, EMERGE peaks were called when the absolute EMERGE prediction read-out exceeded 0.05, with a resolution of a single bp. At this stringency level 83,338 peaks are called genome-wide, covering 2.5% of the human genome. Finally, we applied EMERGE on the variant regions, and identified 1750 peaks that are potential REs (Table 1, Supplementary Data 4). We then intersected atrial CM-enriched ATAC-seq data, putative heart enhancers mapped by EMERGE and variant regions (Supplementary Data 5). On average there were 11.6 ATAC-seq peaks and 16.8 EMERGE peaks per variant region (Table 1) representing potential REs in these loci. We found enrichment of binding motifs for cardiac TFs such as MEF2s, MEIS1 and TBX3/5, as well as for CTCF in these ATAC-seq peaks (Fig. 6d).

Next we determined which of the associated variants in each variant region lie within cardiac REs. It was shown, however, that sub-threshold SNPs with $p < 1 \times 10^{-4}$ that overlap epigenetic RE marks can affect RE activity and are highly likely to represent true disease risk loci[40]. Therefore, to increase discovery rates in the variant regions, we included all subthreshold variants within the variant region with $p < 1 \times 10^{-4}$ (18,807 in total) (Supplementary Fig. 1b, Supplementary Fig. 2b). To assess the number of variants that might interfere with RE function, we determined the tag-counts of both ATAC-seq and EMERGE on all 18,807 SNPs. This resulted in 876 variants (4.6%) with overlapping ATAC-seq and EMERGE signals (Fig. 6e, g Supplementary Data 5). Our approach reduces the number of initial candidates to 16% of the potential causative variants in this locus. We intersected the variants with promoters (−1500 to +500 bp of transcription start

site (TSS)), showing that 309 out of 2005 variants with either ATAC-seq or EMERGE signal lie in promoters (15%) (Fig. 6e, g Supplementary Data 6). To validate our approach to identifying potential variant REs, we performed a luciferase assay on both alleles of 11 potential variant RE in the rat atrial CM cell-line, iAM1[60]. Three of eleven variants tested showed allele-specific regulatory potential (Fig. 6f).

We next listed the potential variant REs that have a PCHi-C signal implicating a potential target gene. For each of the potential variant REs, we determined interactions with our 264 determined causative genes (with score > 11) using PCHi-C data[36]. This resulted in a shortlist of 629 variants that (1) lie in a potential RE and (2) show interaction with one or multiple promoters of genes that are likely target genes of AF (Supplementary Data 6). Moreover, we determined which variants lie in a selection of (cardiac) TF binding motifs, and could therefore change these motifs and influence RE function. Out of the variants located in TF binding motifs, we selected only the ones that also lie in accessible chromatin (ATAC-seq peak). This analysis resulted in the identification of 129 variants located within a TF binding motif present in accessible chromatin (Supplementary Data 7). Out of these variants, PCHi-C shows that 42 are in contact with one or multiple promoters of our AF target genes.

## Discussion

Over 100 loci in the genome are currently known to be associated with AF[12]. Because the variants in these loci are mostly located in non-coding regions of the genome, the prevailing hypothesis is that they interfere with the function of REs that control the transcription patterns and levels of their target genes[61]. Identification of these target genes, however, has proven difficult. In this study, we developed an approach that integrates epigenetic, chromatin conformation and expression data to find probable target genes and variants REs of AF associated loci.

Our approach to define candidate variant RE-target genes involved in AF is based on a number of assumptions (Supplementary Fig. 1). Based on these assumptions we leveraged published Hi-C and PCHi-C data to determined which genes lie in the AF-associated TADs, and which promoters are in contact with the variant regions in relevant tissue[36]. In these data sets weaker or more transient interactions could be missed. Moreover, RE-promoter close proximity does not predict a functional interaction. To address this issue, we generated five different atrial gene expression datasets for target gene analysis, ensuring atrial-expressed genes are included. Therefore, this approach focusses on atrial-driven mechanism underlying AF, not taking into account extra-cardiac mechanisms such as hypertension. A large fraction of contributing genes found in families with AF have a well-established role in cardiac or CM function, and are involved in cardiogenesis (*PITX2*, *TBX5*), cell coupling (*GJA1*, *CAV1*) or ion handling (*KCNN3*, *HCN4*)[62], providing justification for this focus. We generated expression profiles of atrial dorsal wall (non-auricle) tissue as opposed to the often used and more readily available appendage, as this is the most common site of AF initiation, which is thus potentially more relevant for AF etiology[43]. Furthermore, because there is evidence of the involvement of developmental processes in the contribution to the risk of AF[13,17], we include expression data of human fetal atria to detect potential target genes expressed during development.

This approach is useful for identifying potential target genes of loci in which no AF-associated genes can be identified via, e.g., eQTL. Our scoring clearly implicates *SPATS2L* as the potential target gene of the variant region of this locus (Fig. 3e). The role of *SPATS2L* in the heart is unknown, but this locus has been

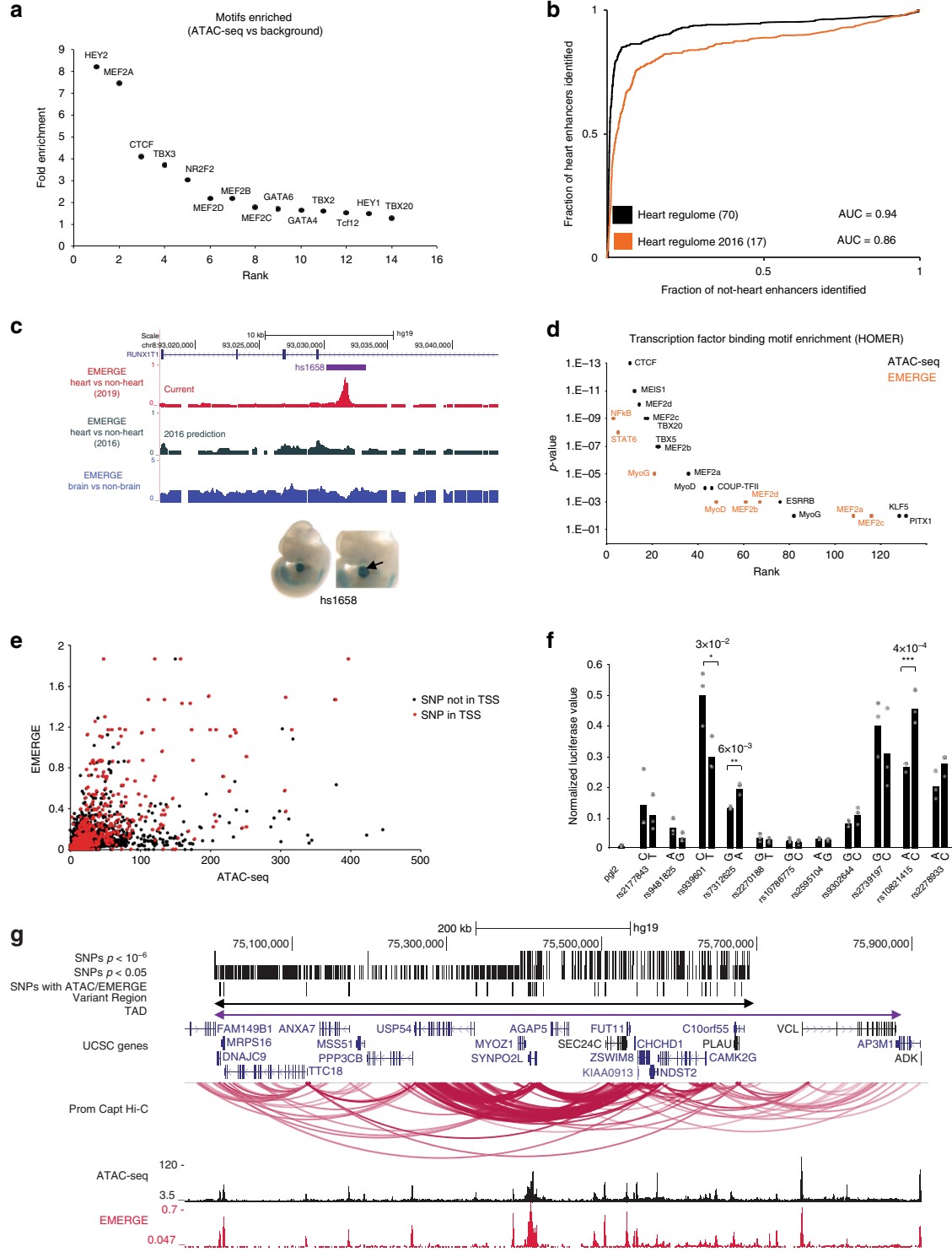

**Fig. 6** Identification of cardiac regulatory elements and functional AF-associated non-coding variants. **a** ATAC-seq motif enrichment, **b** Area under the receiver operating characteristic curves showing the improved prediction of the revised EMERGE, **c** Browser view of human heart enhancer prediction for brain, previous heart prediction[38] and current heart prediction. hs1658 drives LacZ expression in the heart in the mouse embryo (Vista enhancer browser). **d** HOMER motif analysis on peak-calling ATAC-seq, EMERGE and overlapping peaks. **e** ATAC-seq and EMERGE peaks overlapping known AF GWAS variants, as well as variants that lie in promoters (red). **f** Allele-specific luciferase assay performed in iAM-1 cells. Error bars represent SD, *$p < 0.05$, **$p < 0.01$, and ***$p < 0.001$ (two-tailed Student's $t$-test). Source data are provided as Source Data file. **g** SYNPO2L locus with GWAS variants, PCHi-C, ATAC-seq and EMERGE tracks. PCHi-C promoter capture Hi-C, ATAC-seq assay for transposase-accessible chromatin sequencing

implicated in a GWAS for QT duration[63] as well as in a GWAS for bronchodilator response in asthma patients[64]. In this latter study, *SPATS2L* was implicated as a negative regulator of β2-adrenergic receptor levels. β-blockers are known to prevent AF and convert it to sinus rhythm, as well as causing pharmacological remodeling with lasting anti-adrenergic beneficial effects on rhythm upon long-term use (reviewed by ref. [65]). This could point towards a role for *SPATS2L* in conduction, contraction and heart rate in AF mediated through the same adrenergic mechanisms. Other examples include *VCL*, which lies in the rs6480708-associated region (10q22.2), and is one of the highest scoring genes without AF-associated eQTL. *VCL* is essential for stabilization of gap junctions in CMs, involved in anchoring F-actin to the membrane, and has been associated with cardiomyopathy[66,67]. Furthermore, it has a functional interaction with Nav1.5, through which it could contribute to fatal arrhythmia underlying sudden unexplained nocturnal death syndrome[66]. Its role in AF requires further research.

To provide proof of principle for our approach to identify variant region target genes, we deleted parts of variant regions in mice using CRISPR-Cas9. We found that a 40 Kb region homologous to human variant region 680 Kb downstream of *GJA1* acts as a distal cardiac-specific enhancer of *Gja1*. The deleted region contained AF-associated variants as well as epigenetic marks such as ATAC-seq and EMERGE signals indicating cardiac regulatory potential. Deletion of part of the variant region homologue in mice led to a small but significant reduction in expression of *Gja1* (20%), the most likely target gene in our scoring approach, but not of other genes in the TAD. This example suggests our approach can accurately predict target genes. Reduced Cx43 expression in mice does lead to increased fibrosis during aging and is arrhythmogenic[68], suggesting that this could be one of the mechanisms through which modulated expression through this variant region adds to the risk of AF.

We analyzed the ATAC-seq dataset for motif enrichment, and found that it is highly enriched for (atrial) cardiac TFs such as HEY, TBX, and MEF2 family members, COUP-TF2, which is the principal TF for atrial specification in mouse[69], and for CTCF-binding sites. Of 1545 AF-associated SNPs that colocalize with an ATAC-seq called peak, we found 129 AF variants that disrupt selected cardiac TF binding motifs. One of these variants, rs281766, lies in a TBX5 binding site in the 5′ UTR of C2orf47 in the locus of *SPATS2L* (Fig. 3b). The region is highly conserved in mice, and published murine ChIP-seq datasets show Tbx5 binding at this homologous region[70]. The minor allele of variant rs281766 (Fig. 3c) changes the DNA in such a way that it represents the most common position weight matrix binding site of the TBX TF family, which could greatly increase the binding affinity of TBX TFs to its binding motifs (Fig. 3d)[71]. Furthermore, this variant shows interaction with the promoter of the potential target gene *SPATS2L* (Fig. 3b), making this the likely target gene of this potential variant RE. Even though the dynamics of TF binding are dependent on many different factors such as co-factors, DNA conformation and protein concentration, this example illustrates that our method can pinpoint interesting TF-binding site altering variants, and can help generate hypotheses for further study mechanisms through which they exert their effect.

Less than 10% of over 18,000 subthreshold AF-associated SNPs in the variant regions colocalized with REs predicted either on basis of human atrial CM ATAC-seq or on EMERGE, which performance was greatly improved (Fig. 6b). When using both these datasets for selecting candidate variants, this approach further reduces the number of initial candidates to 4.6%. Whether and to what extent our approach misses REs, such as repressors or other functional elements, or incorrectly assigns RE function,

remains to be tested. Although the prioritization approach identifies variants that lie within REs, a variant may not affect the function of an RE. We found that three of 11 fragments tested showed allele-specific activity using a luciferase reporter assay in an atrial CM line[60]. These data suggest that our approach is able to prioritize potentially functional variants. Nevertheless, the involvement of such variant REs in AF-relevant gene regulation requires additional studies.

## Methods

**Ethical statement.** George Washington University IRB approved this protocol. Fetal cardiac tissue from elective abortions was used after written (parental) informed consent was provided. Collection and use of human fetal tissue for research was approved by the Medical Ethics committee of the LUMC (P08.087). De-identified adult cardiac atrial tissue was provided by Washington Regional Transplant Community from non-diseased donor hearts which were not used for transplantation.

**Published datasets used.** For an overview of datasets and analyses, see Supplementary Fig. 2. AF-associated genetic variants were selected from a recent extensive AF GWAS study[12]. Published PCHi-C data from hiPSC-CMs was used[36].

**Nuclei isolation.** Nuclear isolation was performed as previously published[72] using an adapted protocol. Samples were kept at 4 °C throughout the procedure. In short, approximately 100 mg tissue was trimmed to 2 mm pieces, and homogenized in lysis buffer containing RNase inhibitor for 10 s at maximum speed using an Ultra-Turrax homogenizer. Samples were homogenized further using a douncer with loose pestle (10 strokes). After a 10-min incubation in the lysis buffer, 10 strokes were performed with a tight pestle. The lysis procedure was monitored by light microscopy to ensure complete tissue and cell lysis and efficient nuclear extraction. The crude lysate was consecutively passed through mesh filters with pore size of 100 and 30 μm. The final lysate was spun at $1000 \times g$ for 5 min and the resulting pellet was resuspended in 500 μl staining buffer (5% BSA in PBS) supplemented with RNAse inhibitor.

**RNA-seq.** Human adult and fetal atrial free wall tissue were obtained in strict accordance with the tissue regulation of the George Washington University and Leiden University Medical Center. Tissue was stored in RNAlater for whole RNA isolation, or snapfrozen for nuclear isolation, and stored at −80 °C.

Total RNA isolation was performed on approximately 100 mg atrial free wall. In short, the tissue was homogenized using an ultra-Thurrax homogenizer. Whole tissue RNA was isolated using the Trizol method, followed by a DNase treatment and purification using RNeasy MinElute kit (Qiagen). RNA quality, concentration and RNA integrity number (RIN) scores were assessed on the 2100 Bioanalyzer (Agilent Technologies). Samples with RIN scores above 6.8 were selected for RNA-seq.

To obtain CM-specific transcriptomes, nuclei were isolated from snapfrozen tissue samples as described before[50] and incubated with rabbit polyclonal antibodies specific for pericentriolar material 1 (PCM1) (Sigma-Aldrich; HPA023370) at a dilution of 1:400 for 1 h rotating at 4 °C. Next, Alexa Fluor 647-conjugated donkey-anti-rabbit 647 antibodies (ThermoFisher Scientific; 1:500 dilution), and DAPI (1:1000 dilution) were added and the incubation was continued for another hour. Samples were spun at $1000 \times g$ for 10 min and washed with 500 μl staining buffer (5% BSA in PBS) before resuspension in 500 μl staining buffer supplemented with RNAse inhibitor. Intact CM nuclei were sorted on a BD Influx FACS on the basis of DAPI and Alexa Fluor 647 positivity (Supplementary Fig. 9) into the RLT buffer of the RNeasy Plus Micro kit (Qiagen), and RNA was isolated after passing the sample through the gDNA column. RNA yield and purity was assessed using an Agilent 2100 Bioanalyzer in combination with the RNA Pico chips.

**Library preparation and sequencing.** For adult left and right atria, whole transcriptome amplification was performed using NuGEN's Ovation RNA-Seq V2 kit (San Carlos, CA) according to manufacturer's instructions. For fetal left and right atria, Truseq Stranded Total RNA Library Preparation kit (Illumina) was used according to manufacturer's instructions (with 15 PCR cycles). For whole tissue, 50 ng of RNA was used as input, for RNA isolated from CM nuclei approximately 600 pg was used as input. 500 ng cDNA was fragmented to 200–400 bp using Covaris S2 Focused ultrasonicator (Woburn, MA) and the fragmentation parameters described in the NuGEN ENCORE NGS library preparation protocol. Library preparation was done with Ovation Ultralow System V2 using 10 pg–100 ng of fragmented double-stranded DNA. Sequencing was performed on an Illumina Hiseq4000 sequencing system (50-bp single reads).

**Differential expression analysis.** Reads were mapped to hg19 build of the human transcriptome using STAR[73]. Differential expression analysis was performed using the DESeq2 package based on a model using the negative binomial distribution[74].

The false discovery rate (FDR) method of Benjamini-Hochberg ($p < 0.05$) was used to correct $p$-values for multiple testing.

Unsupervised hierarchical clustering was performed on genes differentially expressed using the R package pheatmap, version 1.0.8. (http://cran.r-project.org/web/packages/pheatmap/index.html). DAVID[75] was used to find overrepresented gene ontology (GO) terms and Kegg pathways in the categories 'biological process' and 'molecular function'. Benjamini–Hochberg correction was performed for multiple testing-controlled $p$ values. Significantly enriched terms were functionally grouped and visualized. The highest significant term of the group was displayed as leading term. Datasets were normalized to reference genes expressed in all datasets (RPL32, RPL4, and H2AF2[52]), after which expression was set per dataset at the median, and high expression at the third quartile to enable categorization of expression for target gene identification. HOMER analysis was performed as described[76].

**AF target gene score calculation**. The different datasets were joined using Galaxy (https://usegalaxy.org/). For each gene, the AF target gene final score per locus was calculated for all genes 1 Mb up- and downstream of the lead SNP. For these genes we summarized the following: TAD 2× (present in the same TAD as the variant region or not, score 1 or 0) + PCHi-C 3 × (signal between 0 and 1) + expression adult human whole tissue atria (left and right) (signal between 0-2) + expression adult human left atrial CMs (signal between 0 and 2) + expression fetal human whole tissue atria (left and right) (signal between 0 and 2) + (4 x summary of eQTL [1 = in 1 study, 2 = multiple studies]) (Table 1).

**Animals**. Lines were maintained on an FVB/NRj background. Both male and female animals were used in this study, aged embryonic to 16 weeks of age. Housing, husbandry and all animal care and experimental protocols for mice in this study were conform to the Directive 2010/63/EU of the European Parliament, and was approved by the Animal Experimental Committee of the Academic Medical Center, Amsterdam, and was carried out in compliance with Dutch government guidelines. For tissue harvest, animals were euthanized by 20% $CO_2$ inhalation followed by cervical dislocation for adult heart harvest. Timed pregnant females were euthanized by intraperitoneal injection with a sodium pentobarbital solution, followed by cervical dislocation for embryonic tissue harvest.

**CRISPR/Cas9 genome editing**. Guide RNA (sgRNA) constructs were designed using the online tool ZiFiT Targeter[77]. The sgRNA sequences are summarized in Supplementary Table 5. The Cas9[78] and sgRNA constructs were in vitro transcribed using the MEGAshortscript T7 and mMessage mMachine T7 Transcription Kit (ThermoFisher Scientific). The sgRNAs (10 ng/μl per sgRNA) and Cas9 mRNA (25 ng/μl) were microinjected into the cytoplasm of FVB/NRj zygotes to generate founder mouse lines. Lines were maintained on an FVB/NRj background.

**EMERGE**. Enrichment of putative heart enhancers was tested as predicted by EMERGE. A total of 70 selected functional genomic datasets from a variety of cardiomyocyte (CM) containing tissues, including whole hearts and CMs differentiated from human embryonic and induced pluripotent stem cells (both publicly available and in-house), was integrated using EMERGE. This included ChIP-seq data of enhancer-associated histone methylation marks and TF binding sites and chromatin accessibility data as assessed by DNase1-hypersensitivity and ATAC-seq. Subsequently an overall heart enhancer prediction track was generated by assigning weights to all 70 selected functional genomic datasets through a logistic regression modeling approach that determines a best fit on 84 validated heart enhancers[59].

**ATAC-seq footprinting**. ATAC-seq footprinting was performed as previously described[58].

**qPCR**. Total RNA was isolated from using TRIzol Reagent according to the manufacturer's protocol (Thermo Fisher Scientific). Concentration of RNA was measured using Nanodrop. Genomic DNA was removed using DNase treatment. The Superscript II system (Thermo Fisher Scientific) and oligo dT primers (125 μM) were used to generate cDNA, and expression levels of different genes were assessed with qPCR using the LightCycler 2.0 Real-Time PCR system (Roche Life Science). Primer pairs spanned at least one intron, and they are listed in Supplementary Table 1. The qPCR reaction was performed with LightCycler 480 SYBR Green I Master (Roche), primers (1 μM) and cDNA (equivalent to 5 ng RNA). The amplification protocol consisted of 5 min 95 °C followed by 45 cycles of 10 s 95 °C, 20 s 60 °C, and 20 s 72 °C. Data was analyzed using LinRegPCR[79]. We used two reference genes per experiment based on literature to normalize gene expression (Hprt, Ppia, Rpl4, and Rpl32)[52].

**Cell culture and Luciferase assay**. Primers were designed for selected AF variants to amplify fragments of 400-900 bp (For primers, see Suppl. Table 1) on DNA heterozygous for the selected variant. The pGL2-Promoter Vector (Promega) was linearized with XcmI (New England Biolabs) to create a T-overhang and agarose-gel isolated. PCR products containing both alleles of the selected variants were

ligated using TA-cloning to the pGL2-Promoter Vector. Recombinant plasmid DNA was purified with the PureLink Quick Plasmid Miniprep kit (Thermo Fisher Scientific). Plasmids that were identical except for the different alleles of the variant were selected using Sanger sequencing.

iAM-1 cells were plated at a cell density of $4 \times 10^4$ cells per $cm^2$ and cultured as described[60]. At day 0 of cardiomyogenic differentiation, 500 ng pGL2-Promoter Vector DNA as well as 25 ng pRL-TK Renilla vector (Promega) as normalization control was transfected into the cells in triplo in a 24-well plate using Lipofectamine 3000 (Thermo Fisher Scientific). At day 9 of cardiomyogenic differentiation the cells were harvested. Luciferase measurements were performed on a Promega Turner Biosystems Modulus Multimode Reader luminometer.

**Data processing and statistical analyses**. For data processing of expression data and differential expression analysis, see Supplementary Methods. For both qPCR and luciferase experiments, data was statistically tested using a two-tailed Student's $t$-test.

**Reporting summary**. Further information on research design is available in the Nature Research Reporting Summary linked to this article.

## Data availability
All RNA-seq data have been deposited in the GEO accession GSE127856. Human ATAC sequencing raw sequencing files are available in the NCBI dbGAP database under accession code phs001539.v1.p1. ATAC sequencing post alignment bed format files are available on the Broad Institute's Cardiovascular Disease Knowledge Portal (broadcvdi. org/informational/data) and for direct download using gsutil from gs://cvdi_epigenome/Human/Left_Atrium/. All other relevant data supporting the key findings of this study are available within the article and its Supplementary Information files or from the corresponding author upon reasonable request. Data underlying Figs. 4a–d, 5e–h, 6f and Supplementary Figs. 5c, d, and 6c–g in this study are provided as a Source Data file. A reporting summary for this Article is available as a Supplementary Information file.

## Code availability
Steps undertaken to generate different files using Galaxy are detailed in Supplementary Note 1.

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

## Acknowledgements

The work was supported by Fondation Leducq [to V.M.C., P.T.E., and J.F.M.], and NIH 3OT2OD023848 [to I.R.E.]. This work was supported by grants from the National Institutes of Health (HL127717, HL130804, HL118761 [J.F.M.]; F31HL136065 [M.C.H.]; HL128074 [M.A.N.]; F31 HL137307 [L.E.M.]; Vivian L. Smith Foundation [J.F.M.], State of Texas funding [J.F.M.].

## Author contributions

Conceptualization: V.M.C., A.F.O., F.M.B., P.B., K.V.D. A.F.O. performed adult RNA-seqs, bioinformatics analyses and in vivo mouse experiments. F.M.B. performed in vivo mouse experiments. K.V.D. analyzed ATAC-seq and RNA-seqs, and performed EMERGE re-training. M.C.H., P.E., and J.F.M. performed ATAC-seq. L.M. and M.N. performed and analyzed PCHi-C. K.T.S. and B.J.B. performed fetal RNA-seq. J.L. and A.A.V. performed iAM1 cell culture. M.J.G. provided fetal atrial tissue and contributed to data analysis. IRE provided adult atrial tissue and contributed to data analysis. A.F.O. and V.M.C. wrote manuscript with input from all co-authors.

## Competing interests

P.T.E. is supported by a grant from Bayer AG to Broad Institute focused on the genetics and therapeutics of cardiovascular diseases. P.T.E. has also served on advisory boards or consulted for Bayer AF, Quest Diagnostics and Novartis. J.F.M. is a founder and owns shares in Yap Therapeutics.
