## [Peer Review File · Nature Communications]

Reviewers' comments:

Reviewer #1 (Remarks to the Author):

This is an interesting paper that I regrettably now recognize I do not have the expertise to evaluate. Although well-versed in basic human genetics and the genetics of atrial fibrillation, I do not have the expertise in the specific methodologies used in this analyses to confidently evaluate the methods and results, and conclusions.

I have only brief comments related to the presentation of the paper and otherwise recommend to the editors a more in-depth review of methodology by experts in the specific tools and bioinformatics used in this paper.

Brief comments:

1. The comment in the Abstract stated: "The vast majority of disease-associated genetic variants are thought to alter the functionality of transcription regulatory elements (REs) and target gene expression." , should be adjusted. There are single gene causes of high impact and penetrance and these clearly lead to abnormalities in ion channel function or the action potential duration in the atria. It should be clarified the disease-associated variants referred to in this paper are those from GWAS studies that represent risk alleles or SNPs.

2. In the Intro of the paper it is stated that: 'the mechanisms of AF etiology remain elusive'. This is not true. While the mechanisms of etiology may be heterogeneous, there is significant knowledge of mechanistic etiology based on Mendelian causes of AF and other areas of research.

3. Although I am not expert on the research approach used in this study, I would ask for some detail on the validation of any innovative approaches used.

Reviewer #2 (Remarks to the Author):

Identifying the actual causal genes at GWAS loci is an important undertaking, but can be challenging as the vast majority of the associations reside in non-coding regions. This investigative team elected

to pursue the loci previously reported for atrial fibrillation by both leveraging existing data and generating their own from relevant cell types. This a relatively comprehensive study and has generated a useful resource. These are my concerns:

1. The authors indicate that the "variant regions were on average 300 kbp in length". This is very unclear - why don't they just look for the SNPs in strong LD with a given sentinel? Plus they drop the significance bar to $P=10^{-6}$, another cause for concern. In both instances it would seem that they are introducing a lot of noise in to their analyses - and perhaps the reader can't see the forest for the trees.

2. The authors need to indicate the resolution of the Promoter capture Hi-C - some comparable techniques use a 4 cutter restriction digest step, while others just use a 6 cutter. I know they are using previously published data, but they should indicate here to guide the reader.

3. When the authors sometimes refer to looking 1 Mb in either direction of the sentinel SNP, and at other times they refer to the constraints of the TAD locations, this is very confusing. It is difficult to understand why the authors didn't limit their search for gene(s) within the TAD throughout the study. The jumping back and forth seems inconsistent.

4. The authors indicate that they leveraging existing knowledge on TAD structure. They also need to guide the reader to understand that TAD coordinates are considered largely tissue independent. That said, a recent paper in Cell made some suggestions that may make the community rethink this a little - PMID:30799036

5. The ATAC-seq data is unpublished. It is unclear why this is not being made available to the readership.

6. The manuscript is very long, and the authors should try to eliminate redundancy in the text as much as possible to enable ease of reading.

7. Could the term EMERGE be confusing for the field given the existing Electronic Medical Records and Genomics (eMERGE) Network? In addition, given many investigative groups are now integrating such datasets, it is unclear why this approach needs an acronym at this point in time.

Reviewer #3 (Remarks to the Author):

Comments

I'm not sure your assumptions 2 and 3 are strictly true - are they 'enriched' in close proximity and in atrial tissue expressed genes, rather than all?

Why did you restrict the analysis to 1Mb? You state that TADS are an average of 1Mb, which suggests that a significant minority (up to 50%) are greater than 1Mb. This means you have omitted genes within TADS that are >1Mb away in distance - or have I misinterpreted? If so, this could be excluding many within TAD, interacting genes of interest.

Including SNPs $<10^{-6}$ appears a quite simplistic approach for selecting SNPs, for a project that is so heavily reliant on bioinformatic mapping. Looking for independent signals, utilising credible SNP sets and Bayesian scoring would seem more sophisticated if you want to pinpoint physical regions of the genome that harbour likely functional SNPs.

How did you score 'contact strength'? Is PCHiC data truly quantitative - I thought it was more qualitative?.

Again the eQTL analysis is quite simplistic - any SNP in LD with the focal SNP. Methods such as co-localisation should be employed if bioinformatics is such a central aspect of the manuscript. Strong evidence for disease association, with evidence for eQTL doesn't mean the signals are the same, and the eQTL may well be arising from a weak LD with the true eQTL signal.

I feel reducing the p value to 0.05 of any SNP within an epigenetic mark is far too generous and will inflate false positives. I appreciate the evidence of 'sub-threshold' SNPs, but this is including a large number of SNPs that have been demonstrated by GWAS not to be associated with disease, so almost ignoring GWAS evidence for causal SNPs.

You seem to rely quite heavily on the ATAC-seq data - is this included in the manuscript or not? If so it needs to be presented in full, so the review system can determine its validity, otherwise it should be removed.

Point by point rebuttal for van Ouwerkerk *et al.* NCOMMS-19-05596A.

We would like to thank the reviewers for constructive comments and helpful suggestions. We have addressed all of the points raised in our revised manuscript as detailed below, and feel that these changes have greatly improved the paper.

Reviewer 1

We would like to thank the reviewer for the constructive comments and helpful suggestions.

Brief comments:

1-The comment in the Abstract stated: "The vast majority of disease-associated genetic variants are thought to alter the functionality of transcription regulatory elements (REs) and target gene expression." , should be adjusted. There are single gene causes of high impact and penetrance and these clearly lead to abnormalities in ion channel function or the action potential duration in the atria. It should be clarified the disease-associated variants referred to in this paper are those from GWAS studies that represent risk alleles or SNPs.

We agree with the reviewer that a somewhat more precise phrasing was needed and have made the following change:

Abstract, Page 3, lines 45-47:

"The vast majority of disease-associated genetic variants that lie in non-coding regions found by genome-wide association studies are thought to alter the functionality of transcription regulatory elements (REs) and target gene expression."

2-In the Intro of the paper it is stated that: 'the mechanisms of AF etiology remain elusive'. This is not true. While the mechanisms of etiology may be heterogeneous, there is significant knowledge of mechanistic etiology based on Mendelian causes of AF and other areas of research.

We thank the reviewer for pointing out this error. We made the following adjustment:

Introduction, Page 4, lines 63-65:

"Although for some loci the potential mechanisms through which the AF GWAS loci add to the risk of AF have been identified, the majority have remained largely elusive.¹³"

3- Although I am not expert on the research approach used in this study, I would ask for some detail on the validation of any innovative approaches used.

We use innovative approaches to find variant region-target genes as well as regulatory elements (REs) throughout the variant region that contain AF associated variants. In both approaches we use chromatin contact data (PChi-C). Ultimately we knock out the variant regions containing suspected REs, using CRISPR, to validate the target gene identification.

PChi-C has been published and thoroughly validated, showing that promoter-enhancer interactions are different between cell types and throughout differentiation (Freire-Pritchett *et al.*, 2017; Javierre *et al.*, 2016; Mifsud *et al.*, 2015; Schoenfelder *et al.*, 2015; Siersbaek *et al.*, 2017). This approach has been shown to be useful in the identification of potential target genes of disease-related variants, as was shown in a recent study (Montefiori *et al.*, 2018), where 1999 cardiovascular disease-associated variants were linked using PChi-C to over 300 target genes. Moreover, the variant-target gene interactions identified in this study are enriched for active *cis*-regulatory elements and correspond to gene expression dynamics of cardiac disease relevant genes (Montefiori *et al.*, 2018).

We added the following sentence to the manuscript in order to clarify this point for the reader:

Introduction: page 4 lines 82-84.

“Moreover, recent studies showed that promoter interactions in differentiated cardiomyocytes can identify non-coding DNA that is enriched for active REs that interact with cardiac disease relevant target genes.^{36,37}”

CRISPR-Cas9 has been used previously to delete potentially functional REs and other non-coding regions to gain insight into (human) genome function and gene regulation. For example, super enhancers of *Wap* and α -globin were functionally dissected in vivo in mouse using CRISPR (Hay et al., 2016; Shin et al., 2016), and removal of pairs of limb REs near the same gene lead to a phenotype, while deletion of a single RE did not (Osterwalder et al., 2018). In another study, it was found that distinct human limb malformations are caused by deletions, inversions, or duplications altering the structure of the TAD-spanning *WNT6/IHH/EPHA4/PAX3* locus. Here the authors used CRISPR/Cas genome editing to generate mice with corresponding rearrangements and deletions (Lupianez et al., 2015). Reviews describing these and other examples (Furlong and Levine, 2018; Krijger and de Laat, 2016; Long et al., 2016) have been cited in the revised manuscript. The novelty of our approach is that we deleted mouse genomic sequences homologous to variant regions, possibly containing variant REs, to address their function in gene regulation in vivo.

ATAC-seq identifies accessible chromatin, similar to DNase1 hypersensitivity, and has been well-validated in published studies. We have used this technique to identify accessible sites in cardiomyocyte nuclei of human left atrial tissue, this in itself being innovative. Similarly, the EMERGE tool, used to identify REs, was retrained using more and better input data and a curated set of validated cardiac enhancers as training set. Its high performance has been validated by having it identify validated human cardiac enhancers (ROC curve, Fig. 7B).

Reviewer 2

We would like to thank the reviewer for the constructive comments and helpful suggestions.

1- The authors indicate that the "variant regions were on average 300 kbp in length". This is very unclear - why don't they just look for the SNPs in strong LD with a given sentinel? Plus they drop the significance bar to $P=10^{-6}$, another cause for concern. In both instances it would seem that they are introducing a lot of noise in to their analyses - and perhaps the reader can't see the forest for the trees.

We thank the reviewer for pointing out this concern. We used the $p < 10^{-6}$ as a less stringent threshold around the sentinel SNPs to include subthreshold regions that could hold valuable biologically relevant information pointing towards potential target genes (Wang et al., 2016).

To address the reviewer's concern, we provide here a short overview of an analysis of the differences in determining variant regions using different stringencies of significance as well as LD data. We analyzed the variant region sizes with both $p < 5 \times 10^{-8}$ and $p < 1 \times 10^{-8}$, broadly accepted cut offs for GWAS data (Fadista et al., 2016; Roselli et al., 2018), in order to compare these boundaries with the $p < 10^{-6}$ used in this study. Variant region is defined by the first to the last variant with the chosen p-values to indicate the stretch of DNA with high association (similar to a regional plot). Comparing the previously used $p < 10^{-6}$ with the new $p < 5 \times 10^{-8}$ and $p < 1 \times 10^{-8}$, we find the following differences:

	Original	bp change compared to $p < 1 \times 10^{-6}$	
	$p < 1 \times 10^{-6}$	$p < 5 \times 10^{-8}$	$p < 1 \times 10^{-8}$
Change in total VR length (bp)	30,990,268 bp (total VR length)	7,742,771	13,820,293
# Loci not changed	-	14	12
# Loci absent	-	2	10

This table shows that 14 and 12 loci, respectively, remain unchanged by the significance cutoff. We also see that 2 and 10 loci, respectively, are completely lost due to the increase in significance threshold. Furthermore, we analyzed the difference in variant region size when we take into account the linkage disequilibrium (LD) of the sentinel SNPs. This analysis showed that variants in high LD ($R^2 > 0.8$) as well as the variants in lower LD ($R^2 > 0.2$) of lead variants almost invariably do not span the same coordinates as the genome-wide significant variants, showing that the regions of association with AF do not necessarily coincide with LD regions based on lead variants (Supplementary Figure 2). The variants in LD of both R^2 values in most cases only span a fraction of the regions as determined by the p-value thresholds of the GWAS signal taken from (Roselli et al., 2018).

Therefore we chose to do our analyses by first selecting the genome wide significant variants per locus (with $p < 1 \times 10^{-8}$). By using this significance as threshold we focus on regions with highly significant AF-association. Because our goal is finding functionally relevant regions, we chose to widen the significant regions slightly to ensure inclusion of potential functionally relevant regions at the margins of each locus (which hold (sub-threshold) SNPs in LD with associated variants). We did this by marking the last $p < 10^{-6}$ variant at either side of the $p < 10^{-8}$ boundaries, leading to a modest increase in size of the variant regions of on average 69 kb on each side (22%). We summarize the variant region length of the two additional thresholds in Table 1.

To determine whether the variant region size has an effect on the target gene analysis, we re-analyzing the PChi-C data per new variant region to include 2 additional variant region sizes based on p-value

($p < 5 \times 10^{-8}$ and $p < 1 \times 10^{-8}$). When we implement the new significance thresholds the result is 216 target genes ($p < 10^{-6}$), 203 ($p < 5 \times 10^{-8}$) or 190 ($p < 1 \times 10^{-8}$) target genes. These results have been added to Suppl. Table 2. We concluded that the target gene analysis is changed by 26 genes with the different significance thresholds of the two extremes.

Another consideration is that if we were to adhere to significance $p < 10^{-8}$, we would lose 10 loci in which there are no variants below the given threshold. In order not to exclude these loci, as well as keep the analysis consistent between loci, we therefore retain our $p < 10^{-6}$ threshold.

We added Suppl. Fig. 2 (variant region analysis comparing GWAS association and LD blocks), Suppl. Table 2 (target gene scoring when using two additional thresholds for variant region size) and added the following text to the manuscript:

Results; page 6, lines 104-114:

“Because any AF-associated variant close to the lead SNP could be affecting RE function, we set out to define the genomic range of each AF-associated variant region. We tested various different criteria (SNPs in LD, association p-value cut offs) to define the variant regions (Suppl. Fig. 2). Using the widely accepted genome-wide threshold of $p < 1 \times 10^{-8}$ for AF association of variants resulted in variant regions of high confidence and practical sizes.^{39,12} These regions most likely contain the variant REs that govern candidate target gene expression. Sub-threshold SNPs (association $p < 1 \times 10^{-4}$) can affect RE activity and are highly likely to represent true disease risk loci,⁴⁰ which may be excluded when they flank the variant region borders as defined by these stringent criteria. Therefore, we extended the variant regions in order not to lose potentially functional variants at the margins of each locus by marking the last variant with $p < 1 \times 10^{-6}$ at either side beyond the $p < 1 \times 10^{-8}$ boundaries, leading to a modest increase in variant regions of on average 69 kb (22%) at each boundary (Table 1).”

2. The authors need to indicate the resolution of the Promoter capture Hi-C - some comparable techniques use a 4 cutter restriction digest step, while others just use a 6 cutter. I know they are using previously published data, but they should indicate here to guide the reader.

Indeed, the restriction digest step and the enzyme used for this step are crucial to PChi-C. The data we used (Montefiori et al., 2018) was generated using the 4 bp cutter MboI, which generates fragments with an average size of 422 bp. Using a 4 cutter instead of the more generally used 6 cutter increases specificity of the captured region and increases power to detect short transcription factor binding motifs. This leads to better resolution of the underlying enhancer sequence in the dataset.

We have indicated the resolution in the revised Results section: page 7, lines 124-125.

“The PChi-C dataset was generated using a 4bp cutter, generating fragments of smaller size and increased resolution of underlying RE compared to data generated by a 6bp cutter.”

3. When the authors sometimes refer to looking 1 Mb in either direction of the sentinel SNP, and at other times they refer to the constraints of the TAD locations, this is very confusing. It is difficult to understand why the authors didn't limit their search for gene(s) within the TAD throughout the study. The jumping back and forth seems inconsistent.

We thank the reviewer for this comment. In the revised manuscript we more clearly distinguish between these two parameters. We include both parameters to score candidate variant region target genes because most, but possibly not all, functional RE-target promoter interactions are confined to the same TAD.

Several studies have shown that TADs are the primary units of transcriptional regulation, confining RE-target promoter interactions, and there is ample evidence that disturbance of TAD boundaries can lead to enhancer adoption by genes in neighboring TADs (reviewed in (Furlong and Levine, 2018; Krijger and de Laat, 2016; Long et al., 2016)). Nevertheless, the evidence of cell-to-cell (and even allele-specific) variability in genome architecture and TAD structure, and inter-TAD interactions, suggest that REs could potentially contact promoters outside their TADs (Finn et al., 2019). To include these possible target genes outside variant region TADs, we have increased the region of interrogation around each sentinel SNP to 1.9 Mb in each direction, because a literature search showed that, to date, the largest reported distance between an enhancer and its target gene was 1.9 Mb (Mifsud et al., 2015). In our scoring system, the potential target genes in this region spanning 1.9 Mb up- to downstream of the sentinel SNP gain additional weight when they share the TAD with the variant region.

We adjusted the text to better explain the use of both parameters (1.9 Mb and TAD), include this new selection distance, and implemented the change by re-analyzing all data and correcting all relevant tables.

Results, Page 6, lines 115-121:

“RE-target promoter interactions over distances up 1.9 Mb have been reported.⁴¹ Therefore, to identify genes within reach of potential variant REs, we identified all genes 1.9 Mb upstream to 1.9 Mb downstream of the lead SNPs as the potential target genes per locus (Suppl. Table 2). However, RE activity is largely limited to genes that fall within the same TAD.^{32–34} Since TAD structures are largely tissue independent, we determined the TAD of the variant region by interrogating available Hi-C data for each of the 104 AF-associated loci in our study.⁴² We indicated for each gene 1.9 Mb up- or downstream of the lead variant whether or not it was within the same TAD as the variant region (Suppl. Table 2). ”

4. The authors indicate that they leveraging existing knowledge on TAD structure. They also need to guide the reader to understand that TAD coordinates are considered largely tissue independent. That said, a recent paper in Cell made some suggestions that may make the community rethink this a little - PMID:30799036

We thank the reviewer for this suggestion. Indeed, TAD structures have been found to be largely (about 70%) tissue independent (Dixon et al., 2012). Therefore we chose a dataset of a lymphoblastoid cell line with the highest resolution (Rao et al., 2014) to find the likely TADs within which there is a higher chance of enhancer-promoter interaction. Additionally, it was found by PChI-C in differentiated cardiomyocytes that 89% of variant-target gene interactions are located within the same TAD (Montefiori et al., 2018).

Indeed, the study mentioned by the reviewer shows that TADs are variable between individual cells, and that interactions of two alleles in the same nucleus are independent of each other (Finn et al., 2019). We included reference to this work in the revised manuscript (see below). Nevertheless, in this study, inter-TAD interactions occurred 2 to 3 fold less frequently than intra-TAD interactions, suggesting that there is more frequent interaction between TADs. Moreover, it seems that physical proximity between RE and promoter is needed for transcription (Chen et al., 2018). Transcription-mediated reshaping of 3D genome organization seems to play a role in stabilizing the temporal and spatial aspects required for the transcriptional response. Furthermore, there is ample evidence that RE activity is largely limited to genes that fall within the same TAD, and that disruption of TAD boundaries can cause enhancers to activate genes in neighboring TADs (reviewed in (Furlong and Levine, 2018; Krijger and de Laat, 2016; Long et al., 2016)). Taken together, these data suggest that enhancers likely (but not exclusively) regulate target genes within the same TAD. We take these considerations as support for our approach to include genes

outside the TAD as well as score additionally for presence within the same TAD. The combination of genome organization –by using Hi-C and PCHI-C- and transcription -using RNA-seq- in this study ensure any potential target genes (within and outside of the TAD) are considered that are missed by the other criteria.

To clarify these issues we modified the following sentences:

Introduction page 4, lines 78-87.

“The majority of functional RE-target promoter interactions occur within the same topologically associating domain (TADs);^{29–31} disruption of TAD boundaries can cause REs to activate genes in neighboring TADs (reviewed in ^{32–34}). Nevertheless, TAD structure variability and inter-TAD interactions have been observed as well.³⁵ Moreover, recent studies showed that promoter interactions in differentiated cardiomyocytes can identify non-coding DNA that is enriched for active REs that interact with cardiac disease relevant target genes.^{36,37} However, physical proximity between REs and promoters -as assessed by high resolution chromosome conformation capture technologies³²⁻ is required, but not sufficient, and not all interactions are detected by the conformation capture assays.^{28,32,36} Therefore, determining which specific gene(s) is/are regulated by a variant-affected RE remains a challenge.”

Results, pages 9, lines 175-178.

“We set the threshold above which we consider a gene a potential target gene at 11 in order to remove genes that are expressed (maximal score of 10) but do not score on any of the other criteria. Furthermore, genes that are expressed and show close proximity to the variant region outside of the TAD are included using this score.”

5. The ATAC-seq data is unpublished. It is unclear why this is not being made available to the readership.

A manuscript including the primary ATAC-seq data is currently under review and the data has been deposited on dbGAP, becoming available as soon as this paper has been accepted. The submitted manuscript (Zhang & Hill et al., 2019. Supplemental document 1) has been included in the rebuttal.

6. The manuscript is very long, and the authors should try to eliminate redundancy in the text as much as possible to enable ease of reading.

We have attempted to remove as much redundancy in the results and discussion section as possible, resulting in a reduced and more succinct manuscript of under 5000 words main text, which is a requirement of the journal.

7. Could the term EMERGE be confusing for the field given the existing Electronic Medical Records and Genomics (eMERGE) Network? In addition, given many investigative groups are now integrating such datasets, it is unclear why this approach needs an acronym at this point in time.

Unfortunately we cannot change this acronym, as it is a published program. There should not be any confusion with the following formulation:

Abstract, page 3, lines 52-55.

“We optimized and used EMERGE, a tool for integration of different genomic datasets to predict functional cardiac REs, and accessible chromatin profiles of human atrial cardiomyocytes nuclei to more accurately predict human cardiac REs and identified hundreds of sub-threshold SNPs that colocalize with cardiac REs.”

Reviewer #3

We would like to thank the reviewer for the constructive comments and helpful suggestions.

Comments

1) I'm not sure your assumptions 2 and 3 are strictly true - are they 'enriched' in close proximity and in atrial tissue expressed genes, rather than all?

In assumption 2 we state that the target genes and variant region are more likely to be present in the same TAD. In our scoring of the potential target genes in this region spanning 1.9 Mb up- to 1.9 Mb downstream of the sentinel SNP, we give a higher weight to the genes that lie within the same TAD as the variant region. It has been found by several studies that TADs are the primary units of transcriptional regulation, confining functional RE-target promoter interactions, and there is ample evidence that disturbance of TAD boundaries can lead to enhancer adoption by genes in neighboring TADs (reviewed in ((Furlong and Levine, 2018; Krijger and de Laat, 2016; Long et al., 2016)). Additionally, it was found by PChi-C in differentiated cardiomyocytes that 89% of variant-target gene interactions are located within the same TAD (Montefiori et al., 2018). Nevertheless, the evidence of cell-to-cell (and even allele-specific) variability in genome architecture and TAD structure, and inter-TAD interactions, suggest that REs could potentially contact promoters outside their TADs (Finn et al., 2019).

Concluding, the increased distance of interrogation combined with the added scoring for genes within the TAD allows us to include potential target genes that happen to lie outside the TAD while also including the TAD structure as a factor contributing to the likelihood of a gene being an AF target gene. Therefore we give extra weight to genes within the same TAD. Moreover, we do not disregard genes outside the TAD, by including genes of which the promoters are contacted by a variant region in a neighboring TAD up to 1.9 Mb away. To clarify this we modified the following sentences:

Results, pages 9, lines 175-178.

"We set the threshold above which we consider a gene a potential target gene at 11 in order to remove genes that are expressed (maximal score of 10) but do not score on any of the other criteria. Furthermore, genes that are expressed and show close proximity to the variant region outside of the TAD are included using this score."

Assumption 3 is that REs need to be in close physical proximity to the promoters of their target genes to enable regulation of transcription (reviewed in (Furlong and Levine, 2018; Krijger and de Laat, 2016; Long et al., 2016)).

Regarding assumption 3 we added the following to the manuscript:

Introduction page 4, lines 78-84.

"The majority of functional RE-target promoter interactions occur within the same topologically associating domain (TADs);²⁹⁻³¹ disruption of TAD boundaries can cause REs to activate genes in neighboring TADs (reviewed in ³²⁻³⁴). Nevertheless, TAD structure variability and inter-TAD interactions have been observed as well.³⁵ Moreover, recent studies showed that promoter interactions in differentiated cardiomyocytes can identify non-coding DNA that is enriched for active REs that interact with cardiac disease relevant target genes.^{36,37} "

2) Why did you restrict the analysis to 1Mb? You state that TADS are an average of 1Mb, which suggests that a significant minority (up to 50%) are greater than 1Mb. This means you have omitted genes within

TADS that are >1Mb away in distance - or have I misinterpreted? If so, this could be excluding many within TAD, interacting genes of interest.

We agree that, although among currently known enhancers there are only a few that regulate genes up to 1 Mb away, the analysis in this study might not have the sufficient range. Therefore, we increased the region of interrogation around each sentinel SNP to 1.9Mb in each direction. We chose this distance based on a literature search that showed that the largest reported distance between enhancer-target gene was 1.9 Mb (Mifsud et al., 2015).

Because of this increased genomic region of interrogation we now have 3569 instead of 2073 genes in our analysis.

3) Including SNPs $<10^{-6}$ appears a quite simplistic approach for selecting SNPs, for a project that is so heavily reliant on bioinformatic mapping. Looking for independent signals, utilising credible SNP sets and Bayesian scoring would seem more sophisticated if you want to pinpoint physical regions of the genome that harbour likely functional SNPs.

We thank the reviewer for pointing out this concern. As also discussed in response to question 1 of reviewer 2, we used $p < 10^{-6}$ as a less stringent threshold *around the sentinel SNPs* to include subthreshold regions that could hold valuable biologically relevant information pointing towards potential target genes. It has been shown that active enhancers are enriched for variants with a p-value of between $p < 1 \times 10^{-4}$ and $p < 5 \times 10^{-8}$ in GWAS of disease relevant tissues (Wang et al. 2016).

As detailed in answer to question 1 of reviewer 2, we adjusted and reanalyzed the target gene analysis to include more stringent and widely-used thresholds of $p < 5 \times 10^{-8}$ and $p < 1 \times 10^{-8}$ to determine our SNP set from the GWAS data. A re-evaluation of the GWAS significance (Fadista et al., 2016) confirmed the genome wide significance of $p < 5 \times 10^{-8}$ for common variants (MAF > 5%) in datasets with European populations. The GWAS dataset used here also contains variants with MAF < 5%, and therefore we lower the threshold to $p < 1 \times 10^{-8}$ in the variant enhancer analysis.

Regarding Bayesian scoring: there is insufficient to no experimental data available to apply Bayesian statistics, as this type of analysis requires explicit assumptions about effect sizes at truly associated variants (Stephens and Balding, 2009). This is one of the reasons we are performing this study, identifying relevant variants using different datasets to get a more accurate idea of relevant variants not based on p-value but epigenetic and functional data.

4) How did you score 'contact strength'? Is PCHiC data truly quantitative - I thought it was more qualitative?.

Contact strength was quantified using CHiCAGO, a program that accounts for sequencing capture bias, and that identified interaction frequency reflecting number of reads supporting these interactions (Cairns et al., 2016). Moreover, it focusses on interactions separated by a distance of at least 10kb to exclude random Brownian contacts (Montefiori et al., 2018).

We agree with the reviewer that we do not know whether the number of PCHi-C interactions or PCHi-C strength are correlated with probability of function, although there likely is a relation between proximity and transcription (Chen et al., 2018).

Even though there is evidence that interaction and expression are linked, the dynamic interplay between topology and transcriptional activity has not been elucidated. Therefore, we removed the quantitative analysis from the PCHi-C analysis, and adjusted the analysis to a binary one: any PCHi-C interaction or strength of a promoter with a variant region now is equivalent to a score of 1 instead of a quantitative

score of 1 to 3 based on interaction strength. This analysis led to a reduction in target genes from 337 to 216.

5) Again the eQTL analysis is quite simplistic - any SNP in LD with the focal SNP. Methods such as co-localisation should be employed if bioinformatics is such a central aspect of the manuscript. Strong evidence for disease association, with evidence for eQTL doesn't mean the signals are the same, and the eQTL may well be arising from a weak LD with the true eQTL signal.

We thank the reviewer for the suggestion. We were unclear in our formulation on this point. We merely reported eQTL genes reported by other studies.

We made the following adaptation to the results:

Results, Page 8, lines 159-164:

“Next, we inventoried known eQTL data reported in literature.^{10,12,14-17} In 36 loci, a risk SNP or SNPs in high LD with the risk SNP was associated with variation in gene expression of a gene in human RA appendage, left ventricle or skeletal muscle tissue. eQTL thus indicates an association between risk SNPs and change in expression of a particular gene. Therefore, we included these data in the score calculation of the candidate target gene probability assessment by listing which genes 1.9 Mb up- and downstream of the AF lead SNPs have been linked to AF via eQTL (Suppl. Table 2).”

6) I feel reducing the p value to 0.05 of any SNP within an epigenetic mark is far too generous and will inflate false positives. I appreciate the evidence of 'sub -threshold' SNPs, but this is including a large number of SNPs that have been demonstrated by GWAS not to be associated with disease, so almost ignoring GWAS evidence for causal SNPs.

We agree with the reviewer on this point. Because we limited our search to the region with the highest association –sub-threshold variants within the variant region of high association of $p < 1 \times 10^{-6}$ - we reasoned a p-value threshold of 0.05 would not be introducing too many false positives, especially in combination with the epigenetic datasets used for the analysis (EMERGE and ATAC-seq).

Subthreshold variants ($p < 1 \times 10^{-4}$) with epigenetic marks represent biologically relevant signal, and when these loci are perturbed there are regulatory consequences (Wang et al., 2016). Based on these observations we adjusted our analysis to include only SNPs that lie within the variant region (defined as $p < 10^{-6}$) below a $p < 10^{-4}$.

The changes were applied accordingly in all analyses.

We adjusted the text in the following way: Results pages 13; lines 263-266

“Next we determined which of the associated variants in each variant region lie within cardiac REs. It was shown, however, that sub-threshold SNPs with $p < 1 \times 10^{-4}$ that overlap epigenetic RE marks can affect RE activity and are highly likely to represent true disease risk loci.⁴⁰ Therefore, to increase discovery rates in the variant regions, we included all subthreshold variants within the variant region with $p < 1 \times 10^{-4}$ (18,807 in total) (Fig. 1B, Suppl. Fig. 1B). ”

7) You seem to rely quite heavily on the ATAC-seq data - is this included in the manuscript or not? If so it needs to be presented in full, so the review system can determine it's validity, otherwise it should be removed.

The manuscript containing the ATAC-seq is under review and the data have been deposited on dbGAP. The submitted manuscript has been added to this response for the reviewer's reference (Zhang & Hill et al., 2019. Supplemental document 1).

References

- Cairns, J., Freire-Pritchett, P., Wingett, S. W., Varnai, C., Dimond, A., Plagnol, V., Zerbino, D., Schoenfelder, S., Javierre, B. M., Osborne, C., et al. (2016). CHiCAGO: robust detection of DNA looping interactions in Capture Hi-C data. *Genome biology* **17**, 127.
- Chen, H., Levo, M., Barinov, L., Fujioka, M., Jaynes, J. B. and Gregor, T. (2018). Dynamic interplay between enhancer-promoter topology and gene activity. *Nat Genet* **50**, 1296-1303.
- Dixon, J. R., Selvaraj, S., Yue, F., Kim, A., Li, Y., Shen, Y., Hu, M., Liu, J. S. and Ren, B. (2012). Topological domains in mammalian genomes identified by analysis of chromatin interactions. *Nature* **485**, 376-380.
- Fadista, J., Manning, A. K., Florez, J. C. and Groop, L. (2016). The (in)famous GWAS P-value threshold revisited and updated for low-frequency variants. *Eur J Hum Genet* **24**, 1202-1205.
- Finn, E. H., Pegoraro, G., Brandao, H. B., Valton, A. L., Oomen, M. E., Dekker, J., Mirny, L. and Misteli, T. (2019). Extensive Heterogeneity and Intrinsic Variation in Spatial Genome Organization. *Cell* **176**, 1502-1515 e1510.
- Freire-Pritchett, P., Schoenfelder, S., Varnai, C., Wingett, S. W., Cairns, J., Collier, A. J., Garcia-Vilchez, R., Furlan-Magaril, M., Osborne, C. S., Fraser, P., et al. (2017). Global reorganisation of cis-regulatory units upon lineage commitment of human embryonic stem cells. *Elife* **6**.
- Furlong, E. E. M. and Levine, M. (2018). Developmental enhancers and chromosome topology. *Science* **361**, 1341-1345.
- Hay, D., Hughes, J. R., Babbs, C., Davies, J. O. J., Graham, B. J., Hanssen, L., Kassouf, M. T., Marieke Oudelaar, A. M., Sharpe, J. A., Suci, M. C., et al. (2016). Genetic dissection of the alpha-globin super-enhancer in vivo. *Nat Genet* **48**, 895-903.
- Javierre, B. M., Burren, O. S., Wilder, S. P., Kreuzhuber, R., Hill, S. M., Sewitz, S., Cairns, J., Wingett, S. W., Varnai, C., Thiecke, M. J., et al. (2016). Lineage-Specific Genome Architecture Links Enhancers and Non-coding Disease Variants to Target Gene Promoters. *Cell* **167**, 1369-1384 e1319.
- Krijger, P. H. and de Laat, W. (2016). Regulation of disease-associated gene expression in the 3D genome. *Nat Rev Mol Cell Biol* **17**, 771-782.
- Long, H. K., Prescott, S. L. and Wysocka, J. (2016). Ever-Changing Landscapes: Transcriptional Enhancers in Development and Evolution. *Cell* **167**, 1170-1187.
- Lupianez, D. G., Kraft, K., Heinrich, V., Krawitz, P., Brancati, F., Klopocki, E., Horn, D., Kayserili, H., Opitz, J. M., Laxova, R., et al. (2015). Disruptions of topological chromatin domains cause pathogenic rewiring of gene-enhancer interactions. *Cell* **161**, 1012-1025.
- Mifsud, B., Tavares-Cadete, F., Young, A. N., Sugar, R., Schoenfelder, S., Ferreira, L., Wingett, S. W., Andrews, S., Grey, W., Ewels, P. A., et al. (2015). Mapping long-range promoter contacts in human cells with high-resolution capture Hi-C. *Nat Genet* **47**, 598-606.
- Montefiori, L. E., Sobreira, D. R., Sakabe, N. J., Aneas, I., Joslin, A. C., Hansen, G. T., Bozek, G., Moskowitz, I. P., McNally, E. M. and Nobrega, M. A. (2018). A promoter interaction map for cardiovascular disease genetics. *Elife* **7**.

- Osterwalder, M., Barozzi, I., Tissieres, V., Fukuda-Yuzawa, Y., Mannion, B. J., Afzal, S. Y., Lee, E. A., Zhu, Y., Plajzer-Frick, I., Pickle, C. S., et al. (2018). Enhancer redundancy provides phenotypic robustness in mammalian development. *Nature* **554**, 239-243.
- Rao, S. S., Huntley, M. H., Durand, N. C., Stamenova, E. K., Bochkov, I. D., Robinson, J. T., Sanborn, A. L., Machol, I., Omer, A. D., Lander, E. S., et al. (2014). A 3D map of the human genome at kilobase resolution reveals principles of chromatin looping. *Cell* **159**, 1665-1680.
- Roselli, C., Chaffin, M. D., Weng, L. C., Aeschbacher, S., Ahlberg, G., Albert, C. M., Almgren, P., Alonso, A., Anderson, C. D., Aragam, K. G., et al. (2018). Multi-ethnic genome-wide association study for atrial fibrillation. *Nat Genet* **50**, 1225-1233.
- Schoenfelder, S., Furlan-Magaril, M., Mifsud, B., Tavares-Cadete, F., Sugar, R., Javierre, B. M., Nagano, T., Katsman, Y., Sakthidevi, M., Wingett, S. W., et al. (2015). The pluripotent regulatory circuitry connecting promoters to their long-range interacting elements. *Genome research* **25**, 582-597.
- Shin, H. Y., Willi, M., HyunYoo, K., Zeng, X., Wang, C., Metser, G. and Hennighausen, L. (2016). Hierarchy within the mammary STAT5-driven Wap super-enhancer. *Nat Genet* **48**, 904-911.
- Siersbaek, R., Madsen, J. G. S., Javierre, B. M., Nielsen, R., Bagge, E. K., Cairns, J., Wingett, S. W., Traynor, S., Spivakov, M., Fraser, P., et al. (2017). Dynamic Rewiring of Promoter-Anchored Chromatin Loops during Adipocyte Differentiation. *Mol Cell* **66**, 420-435 e425.
- Stephens, M. and Balding, D. J. (2009). Bayesian statistical methods for genetic association studies. *Nat Rev Genet* **10**, 681-690.
- Wang, X., Tucker, N. R., Rizki, G., Mills, R., Krijger, P. H., de Wit, E., Subramanian, V., Bartell, E., Nguyen, X. X., Ye, J., et al. (2016). Discovery and validation of sub-threshold genome-wide association study loci using epigenomic signatures. *Elife* **5**.

REVIEWERS' COMMENTS:

Reviewer #2 (Remarks to the Author):

The authors have satisfied the concerns of this reviewer